# Resampling of ENSO teleconnections: accounting for cold season evolution reduces uncertainty in the North Atlantic

Martin P. King[1,2,3], Camille Li[2,3], and Stefan Sobolowski[1,3]

[1]NORCE Norwegian Research Centre, Bergen, Norway
[2]Geophysical Institute, University of Bergen, Bergen, Norway
[3]Bjerknes Centre for Climate Research, Bergen, Norway

**Correspondence:** Martin P. King (martin.king@uib.no)

**Abstract.** We re-examine the uncertainty of the El Niño–Southern Oscillation (ENSO) teleconnection to the North Atlantic following the investigation of Deser et al. (2017) (DES2017). Our analyses are performed on the November-December (ND) and January-February (JF) means separately, and for a geographical area that covers a larger extent in the midlatitude North Atlantic than DES2017. The motivation for splitting the cold season in this way arises from the fact that the teleconnection patterns and underlying physical mechanisms are different in late fall compared to mid-winter. As in DES2017, our main technique in quantifying the uncertainty is bootstrap resampling. Amplitudes and spatial correlations of the bootstrap samples are presented together effectively using Taylor diagrams. In addition to the confidence intervals calculated from Student's $t$ tests and the percentiles of anomalous sea-level pressure (SLP) values in the bootstrap samples, we also investigate additional confidence intervals using techniques that are not widely used in climate research, but have different advantages. In contrast to the interpretation by DES2017, our results indicate that we can have confidence (at the 5% significance level) in the patterns of the teleconnected SLP anomalies. The uncertainties in the amplitudes remain large, with the upper-percentile anomalies at up to 2 times those of the lower percentiles in the North Pacific, and 2.8 times in the North Atlantic.

## 1 Introduction

During the 1980's–90's, dynamical understanding of tropical-extratropical atmospheric teleconnections saw substantial progress, with observational analyses beginning in earnest as data from in-situ and satellite measurements became available. Trenberth et al. (1998) provides an excellent review of many foundational studies performed during this period establishing the idea that tropical SST perturbations such as El Niño–Southern Oscillation (ENSO) can drive upper-tropospheric divergence, which in turn acts as a source for planetary scale Rossby waves with far-reaching climate impacts. Since then, a major development was the discovery of ENSO's influences on the stratosphere, and the stratosphere as a pathway to influence the surface climate of the extratropics (reviews by Domeisen et al. 2019; Baldwin et al. 2019). More recent efforts in the field include studies on the effects of ENSO sea-surface temperature (SST) diversity and amplitude, and nonlinearity of the teleconnections (e.g.,

Feng et al. 2017; Frauen et al. 2014; Garfinkel et al. 2013; Toniazzo and Scaife 2006; Trascasa-Castro et al. 2019; Weinberger et al. 2019; Zhang et al. 2018). The role of tropical-extratropical teleconnections in the predictability of Northern Hemisphere climate, and how prediction systems can be improved by including these processes correctly, was recognized early on. Attention to this aspect has intensified in recent years (e.g., Hardiman et al. 2020; Scaife et al. 2014, 2016). However, establishing the teleconnection patterns and amplitudes as well as the physical mechanisms involved remains a challenge owing to large atmospheric internal variability in the extratropics.

The issue of uncertainty in the ENSO teleconnection to the Northern Hemisphere during wintertime (DJF) was raised by Deser et al. (2017) (DES2017 hereafter). They used bootstrapping (random sampling with replacement) to show that the sea level pressure (SLP) composites associated with ENSO vary considerably in both pattern and amplitude, due primarily to sampling fluctuations arising from internal (non-ENSO forced) atmospheric variability rather than ENSO variability, as the ENSO SST composites (average of selected events) are relatively robust and hence have very narrow confidence intervals (see Fig. 5 in DES2017). The North Atlantic circulation response exhibits a large range of amplitudes, different levels of statistical significance, and even anomalies of opposite signs in some locations. In contrast, the North Pacific teleconnection has lower uncertainty in amplitude than the North Atlantic, with a high confidence in sign. Through the use of large multi-model ensembles DES2017 also examined model biases in both the forced response to ENSO and the internal atmospheric variability. A key message from their paper is that even with nearly 100 years of observations, the observed response to ENSO over the North Atlantic sector in boreal winter is very uncertain.

The late-fall to early-winter (Nov-Dec) ENSO teleconnection in the North Atlantic is notably different spatially from its late-winter (Jan-Feb) counterpart (e.g., Moron and Gouirand 2003). The Nov-Dec teleconnection resembles the East Atlantic pattern, while the Jan-Feb teleconnection projects onto the NAO pattern (King et al. 2018a; Mezzina et al. 2020). Consistently, Molteni et al. (2020) examined the PDFs of NAO index obtained through resampling a reanalysis record and found a change from positive to negative (negative to positive) NAO index associated with the warm (cold) ENSO phase during the cold season (although the use of a single index loses some accuracy in the spatial patterns). The fall-to-winter evolution has implications for the underlying teleconnection mechanism and associated impacts. Recently, King et al. (2018a) highlighted these intra-seasonal differences, and suggested that the Nov-Dec ENSO teleconnection to the North Atlantic involves a tropospheric pathway, unlike the Jan-Feb teleconnection which involves both tropospheric and stratospheric pathways (also see Ayarzagüena et al. 2018; Domeisen et al. 2019; Hardiman et al. 2019; Jiménez-Esteve and Domeisen 2018). A statistically significant relationship between ENSO and western European temperatures in Nov-Dec was also presented by King et al. (2018a).

The mechanisms behind and model performance in representing the varying ENSO teleconnection during the extended cold season have been explored. Ayarzagüena et al. (2018) suggest that Rossby wave propagation in the troposphere excited by atmospheric diabatic heating (a result of latent heating related to precipitation) in the western tropical Atlantic is responsible for the Nov-Dec teleconnection. The late fall response may also be modulated by SST forcing from the tropical west Pacific (Bladé et al., 2008) or atmospheric diabatic heating over the Indian Ocean, which is itself amplified by ENSO (Abid et al., 2021; Joshi et al., 2021). A number of studies (Ayarzagüena et al., 2018; Joshi et al., 2021; King et al., 2018a; Molteni et al., 2020) report that models are generally able to simulate the varying teleconnection to the North Atlantic from November through February

in initialized hindcasts, while "free-running" coupled-model experiments have less success, producing an NAO-like pattern similar to the observed late-winter teleconnection through all these months with at best a weak signature of the transition.

In this study, we revisit the question of uncertainty in the ENSO teleconnection to the North Atlantic. The major methodological differences compared to DES2017 are that we consider a varying ENSO teleconnection through the cold season, as well as a larger North Atlantic domain that includes the mid-latitudes, where an important part of the teleconnection signal resides. As in DES2017, our main technique for estimating uncertainty is bootstrap resampling (see Hesterberg (2015) and chapters 10, 11 in Efron and Hastie (2016) for good general introductions). This technique is widely employed in climate

research for estimating uncertainties in teleconnections (e.g., Michel et al. 2020), estimating prediction skill in large ensembles (e.g., Stockdale et al. 2015), and investigating the effects of sampling variability (e.g., Cash and coauthors 2017). Bootstrapping artificially increases the number of samples or sample size. It is simple to implement and it does not make any assumption about the underlying distribution. However, it should not be regarded as a substitute for producing larger model ensembles or making longer observations, because certain properties of the bootstrap distributions (such as the location parameter) are still

dependent on the original samples (Hesterberg, 2015).

       The three subsections in Results correspond to the steps of our investigation: **(a)** Analyze the sampling variability problem of SLP composites associated with ENSO for Jan-Feb and Nov-Dec means separately in addition to DJF means, using a North Atlantic domain that includes the midlatitude region in contrast to the polar area used by DES2017; **(b)** investigate asymmetry in El Niño and La Niña-related SLP composites; and **(c)** investigate several types of confidence intervals that are available

via bootstrapping in addition to the ordinary $t$ intervals. In designing the bootstrap tests with these factors in mind, we aim to present a more nuanced perspective of the ENSO teleconnection to the North Atlantic with reduced uncertainty compared to earlier studies which considered the standard DJF winter season view.

## 2    Data and methods

### 2.1    Data

To be consistent with DES2017, we use the same datasets. The SLP data is from the NOAA-CIRES Twentieth Century Reanalysis V2c (Compo and Coauthors, 2011). The sea surface temperature (SST) used to construct the Nino3.4 index is from HadISST1.1 (Rayner and Coauthors, 2003). DES2017 verified that their results are robust across different datasets; our experience has been similar when dealing with SST, SLP, and geopotential heights in monthly to seasonal means for tropical-extratropical teleconnections (e.g., King et al. 2018a, b).

### 2.2    SLP composites and bootstrapping

El Niño and La Niña events are identified according to when the magnitude of the Nino3.4 index is greater than or equal to one standard deviation. Table 1 lists the selected ENSO events for calculating the DJF, Nov-Dec and Jan-Feb SLP composites, with further explanation given at the beginning of Sect. 3.1. For each category, we have a set of SLP anomaly fields denoted

$\mathbb{E}_o$ or $\mathbb{L}_o$, which are approximations of the populations associated with El Niño and La Niña, respectively. The observed (i.e., 'original' to differentiate it from 'bootstrap') SLP composite $\mathbf{C}_o$ for ENSO is then calculated as $\mathbf{C}_o = \overline{\mathbb{E}_o} - \overline{\mathbb{L}_o}$, where the overbar denotes the mean of the set. In constructing a bootstrap composite $\mathbf{C}^*$, we draw randomly with replacement from the sets of $\mathbb{E}_o$ or $\mathbb{L}_o$ to form $\mathbb{E}^*$ or $\mathbb{L}^*$, respectively; and $\mathbf{C}^* = \overline{\mathbb{E}^*} - \overline{\mathbb{L}^*}$ is then calculated. The sample size of each bootstrap $\mathbb{E}^*$ and $\mathbb{L}^*$ composite is the same as the observed $\mathbb{E}_o$ and $\mathbb{L}_o$ respectively, meaning that the bootstrap composites contain the same number of El Niño or La Niãa events as the observations (except when investigating the effect of sample size in Sect. 3.1). Multiple bootstrap composites are generated in this way to form a set $\mathbb{C}^*$. As in DES2017, we typically create 2000 bootstrap composites (i.e., there are 2000 members in $\mathbb{C}^*$). Larger numbers of bootstrap samples have been tested for some cases and the results are not so different that the findings are affected.

## 2.3 Confidence intervals and significance testing

The ordinary $t$ confidence intervals are written conventionally as $\mu = \overline{x} \pm t \cdot SE$ (Bulmer, 1979), where $\mu$ is the population mean, $\overline{x}$ a sample mean calculated from $n$ elements, and $SE = s/\sqrt{n}$ the standard error estimated from $s$ the sample standard deviation. In the two-tailed $t$-test with the null hypothesis $\mu = 0$, we check if $\mu = |\overline{x}| - t \cdot SE > 0$ is true, in which case we can reject the null hypothesis. The $t$ value is usually obtained from a table (e.g., p. 265 in Spiegel and Liu 1999). For example, for a 5% significance level, we look up $t_{0.025}$ for a two-tailed test – that is, a 5% probability split evenly between two tails, which is one reason that a normal (unskewed) distribution is required for the $t$ test. Rewriting the $t$ test using the notation introduced in the previous paragraph for the observed composite gives $|\mathbf{C}_o| - t \cdot \mathbf{SE}_o > 0$, which is checked gridpoint-wise to see where the null hypothesis can be rejected, and hence the signal can be considered statistically significant. The method described here is widely used in climate research (and also here in Sects. 3.1 and 3.2 to check statistical significance). It is useful to describe this general approach because in Sect. 3.3, we discuss three additional types of confidence interval that bootstrapping allows us to calculate.

## 2.4 Taylor diagram

The number of bootstrap composites in $\mathbb{C}^*$ is rather large, and the composites themselves exhibit varying spatial patterns and amplitudes. An effective way to summarize this information is through the Taylor diagram (Taylor, 2001). A Taylor diagram is plotted in polar coordinates $(r, \alpha)$, where we define the radial distance of each plotted point from the origin as the ratio of the spatial amplitude (Euclidean norm) of a bootstrap composite to that of the observed composite:

$$r = \sqrt{\sum_{i,j} \cos(\phi_j) \mathbf{C}^*(i,j)^2} \Bigg/ \sqrt{\sum_{i,j} cos(\phi_j) \mathbf{C}_o(i,j)^2},$$

and the cosine of the angle $\alpha$ from the positive $x$-axis direction as the spatial correlation between the bootstrap and observed composites:

$$\cos(\alpha) = \sum_{i,j} \cos(\phi_j) \mathbf{C}^*(i,j) \mathbf{C}_o(i,j) \Bigg/ \sqrt{\sum_{i,j} cos(\phi_j) \mathbf{C}^*(i,j)^2} \sqrt{\sum_{i,j} cos(\phi_j) \mathbf{C}_o(i,j)^2}.$$

Both the spatial amplitude and spatial correlation are area-weighted (by the cosine of latitude, $\cos(\phi_j)$ above). The symbol $\sum_{i,j} \equiv \sum_i \sum_j$ represents the summation over grid points in a chosen spatial domain. For the North Pacific analysis, we select the domain 150°E–120°W, 30°–60°N; while for the North Atlantic the domain 50°W–10°E, 20°–60°N is used for Nov-Dec, and 60°W–0°E, 30°–70°N for DJF and Jan-Feb. The exact choices of the boundaries might be somewhat subjective but are chosen to enclose the important anomalies in the observed composites within the geographical area of interest (i.e., North Pacific, North Atlantic).

In practice, we compute $r$ and $cos(\alpha)$ using the GrADS functions `atot` and `scorr`. The general interpretation of the Taylor diagram is that the radial distance of a point indicates how strong the anomalies in a bootstrap composite are compared to the observed composite, and the proximity from the $x$-axis indicates how closely the pattern of a bootstrap composite resembles the observed composite.

## 3   Results

### 3.1   Uncertainties of ENSO teleconnections in Nov/Dec and Jan/Feb

First, the observed SLP composites ($\mathbf{C}_o$ described in Sect. 2) for ENSO are calculated for the period 1920–2013, the same period used in DES2017. In addition to the standard DJF composite, we compute composites for Nov-Dec (ND) and Jan-Feb (JF). The events included in the various seasonal composite calculations in this study are listed in Table 1. Also similar to the DES2017 approach, the Nino3.4 indices for NDJ, ON and DJ are used to identify SLP anomalies for DJF, ND, and JF, respectively. One reason for using this one-month lag is to account for a delay in the influence of tropical SST anomalies on the extratropics through both the tropospheric and stratospheric pathways. It turns out that exactly the same years are selected for NDJ and DJ, and they are also mostly the same events as for ON. For Nino3.4 in NDJ and DJ, our method selects 18 El Niños with 1 event being different to DES2017 (we do not have the 1968/69 event, but instead 1977/78), and 16 La Niñas instead of 14 (we do not have the 1938/39 event, but instead 1950/51, 1970/71, and 2011/12). For the ON Nino3.4, there are 18 and 17 events selected for El Niño and La Niña, respectively. .

The value of separating the cold season into late fall (ND) and winter (JF) has been addressed by previous studies investigating statistical and dynamical properties of ENSO teleconnections to the North Atlantic (Abid et al., 2021; Ayarzagüena et al., 2018; King et al., 2018a, b; Moron and Gouirand, 2003; Toniazzo and Scaife, 2006). Shown in Fig. A1 are composites of the four months individually (also see Moron and Gouirand 2003 in particular) which indicate the differences qualitatively and confirm that it is reasonable to perform analyses on ND and JF means. The DJF SLP composite as well as the spatial pattern of significant anomalies (at the 5% significance level) are very similar to those presented by DES2017 (Fig. 1a). There is a strengthening of the Aleutian Low with amplitude up to about 8 hPa; in the North Atlantic the anomalies project onto the negative NAO, with about 4 hPa weakening of the southern center (Azores High) and up to 3 hPa weakening of the northern center (Icelandic Low, but with a very small area of statistical significance). Panel c shows the composite for JF. The main difference compared to DJF is that the anomaly in the northern center in the Atlantic sector strengthens to 4 hPa with a larger area of statistical significance. The composite for ND (panel b) instead resembles the East Atlantic pattern.

Next, bootstrapping is carried out to produce resampled ENSO composites as described in Sect. 2. The corresponding Taylor diagrams for the North Atlantic sector are shown in the top row of Fig. 2. The large black dot on the $x$-axis at coordinates ($r = 1$, $\alpha = \arccos 1$) indicates the observed composite $\mathbf{C}_o$, and the small dots represent the 2000 bootstrap composites in $\mathbb{C}^*$. The blue and red lines indicate the 5th and 95th percentiles of the amplitude ratios or spatial correlations: we refer to these as the *bootstrap percentiles* bracketing the *bootstrap confidence interval*, which contains 90% of the bootstrap composites (see also caption). The second and third rows in Fig. 2 show the spatial pattern of bootstrap composites close to the confidence interval bounds in both the amplitude ratio *and* spatial correlation (i.e., the point closest to the intersection of the blue arc and straight blue line, or of the red arc and red straight red line).

According to the Taylor diagrams, separating the cold season into ND and JF reduces the uncertainty in the ENSO tele-connection to the North Atlantic. Specifically, the cloud of bootstrap composites becomes denser, contracting towards the $r = 1$ semi-circle and the $x$-axis. In terms of the amplitude ratios, the bootstrap confidence intervals for DJF, ND and JF are (0.57,1.59), (0.69,1.56) and (0.58,1.57) respectively, meaning that the upper bounds are about 2.8, 2.3, and 2.7 times the lower ones. In terms of spatial correlations, the improvements are clear for both ND and JF over the traditional DJF winter season definition. Specifically, the bootstrap confidence intervals are narrower and the 5th percentiles are larger, sitting close to 0.8 (straight blue lines in panels b, c) compared to 0.69 for the DJF bootstrap distribution. As a visual guide for the comparison, the thick gray arc indicating the confidence interval for DJF in panel a is repeated in panels b and c. Even for DJF, our analysis, which includes a larger mid-latitude North Atlantic domain than in DES2017, indicates some certainty in the spatial pattern because the confidence interval spans only positive spatial correlations (0.69 to 0.99 given in Fig. 2a). If we instead construct a DJF Taylor diagram for a "polar" North Atlantic area (poleward of $60°$N) similar to DES2017, the uncertainty is larger in both the spatial correlations and amplitude ratios (Fig. 3a). The 5th to 95th percentile interval contains bootstrap composites at the lower limit that have limited areas of statistical significance (Fig. 3b); the midlatitude region that exhibits larger patches of statistically significant SLP anomalies (first column in Fig. 2) are not included, and hence, cannot contribute to narrowing the uncertainty.

The spatial patterns of the bootstrap percentiles in Fig. 2 are consistent with the $t$ test results shown in gray shading (Fig. 1), indicating robust signals where the confidence interval is bounded by significant anomalies of the same sign. For DJF, the southern center has statistically significant negative anomalies at both the 5th and 95th bootstrap percentiles, while the northern center has a very small area of statistical significance at the lower 5th percentile (comparing panels d, g). For ND and JF, both the southern and northern centers are bounded by bootstrap composites showing the same (expected) sign. To summarize, these results show that separating the DJF winter into JF and ND reduces the uncertainty in the pattern of the ENSO teleconnection (as indicated by the narrower intervals for spatial correlations in panels b, c), with a slight improvement in the certainty of the amplitude for ND only. This is examined further in sections 3.2 and 3.3.

DES2017 used data in 1920–2013 to coincide with the availability of model data they also analyzed. We have checked the longer period of 1870–2014 available in the reanalysis data (ENSO events selected are listed in Table 1). The result is shown in Fig. A2, with the top row being the observed composites, and the bottom row the corresponding Taylor diagrams. The main difference compared to Figs. 1, 2 is that for the JF analysis, the bootstrap confidence intervals are narrower for both amplitude

ratio and spatial correlation, with the 5th percentile for the latter increasing from 0.77 to 0.84. Otherwise, using the longer reanalysis period with more available events for the resampling does not produce markedly different results that affect the conclusions.

The effect of sample size (number of ENSO events) on the robustness and uncertainty range of ENSO signals is often estimated using sub-sampling techniques, especially in large ensembles of modelling experiments (e.g., Michel et al. 2020; Weinberger et al. 2019). Although the total number of available observed events is greatly limited compared to a large model ensemble, we can still examine the effect of sample size on the uncertainty by sub-sampling with replacement. We perform this test using sample sizes of 5, 10, 15, ..., etc. for the cases in Fig. A2 (see Fig. A3). As expected, the teleconnection

becomes more robust with sample size: the uncertainty range for amplitude decreases and the spatial correlations increases as the number of events increases from 5 to 25. At spatial correlations of about 0.6, the bootstrap composites begin to consistently have similar structures to the observed ones. Using this information, our estimate of the minimum sample sizes required for robust signals are 20, 15, and 15 for DJF, JF and ND, respectively (panels j, h, i in Fig. A3, marked with *). This result is consistent with the estimates of Weinberger et al. (2019) for a 41-member model ensemble, even given differences in terms

of ENSO definitions, variables, months and regions analysed. As the sample size continues to increase beyond the original observed number of events, the confidence intervals narrow further and the bootstrap composites, unsurprisingly, converge to the observed composites (not shown). Note that Taylor diagrams provide only a general diagnostic for this sample size test; the spatial patterns of the bootstrap composites at the lower limits have to be examined visually (as in the second row of Fig. 2) to provide more details.

While the focus of this study is the North Atlantic, we briefly touch on ENSO's effects in the North Pacific. The analyses in Fig. 2 are repeated for the North Pacific, and shown in Fig. 4. In general, the North Pacific is less affected by sampling variability in both the amplitude ratios and spatial correlations. The teleconnection pattern is consistent across all three seasonal means, unlike the North Atlantic, where ND is different. The clouds of bootstrap composites are tighter and closer to the $x$-axis, with less uncertainty for DJF than JF or ND. For JF, the bootstrap confidence interval for the amplitude ratios is (0.75,1.31), where

the upper bound is 1.75 times the lower one (close to the factor of 2 reported by DES2017). The uncertainty in the amplitude can be important for climate impacts assessment (Deser et al. 2018; Michel et al. 2020).

Another resampling technique is known as the permutation test. We have performed this test with an aim to demonstrate its workings in quantifying uncertainty of the systematic difference between North Atlantic SLP anomalies associated with El Niño and La Niña. This is essentially equivalent to the above assessments which examine the hypothesis of El Niño minus La

Niño composites are significantly different from zero. The result is reported in Appendix A.

## 3.2   Uncertainties in El Niño and La Niña teleconnections

The aim of this subsection is to examine the uncertainties for SLP anomalies associated with El Niño and La Niña separately (based on Niño3.4 as above). Additional analyses also allow us to inspect whether there is any asymmetry in the anomaly patterns associated with these teleconnections. .

The observed composites are shown in the top row of Fig. 5, and the corresponding Taylor diagrams for the North Atlantic are shown in the bottom row. These Taylor diagrams do not indicate any serious problem with the uncertainty in the observed composite spatial patterns as even the 5th percentiles of the bootstrap spatial correlations are quite high at 0.59 and higher. Furthermore, as also reported by DES2017 for DJF, we find that the SLP anomalies during El Niño and La Niña events in the ND and JF seasons do not indicate any asymmetry in terms of the sign of the anomalies over the domains used in the

Taylor diagrams; this is true for both the North Atlantic and North Pacific sectors. However, the spatial extents of significant teleconnections in the North Atlantic are different. During El Niño for ND and La Niña for JF, the SLP anomalies in the North Atlantic have larger areas of statistical significance (panels a, d compared with b, c), as well as narrower bootstrap confidence intervals in the spatial correlations (panels e, h compared with f, g). Note also the smaller scatter of the bootstrap composites and their proximity to the observed composite on the $x$ axis in panels e, h. Without further study and more observations, it is not

possible to provide a physical explanation for the differences in the uncertainties between these composites (but see Hardiman et al. 2019).

       We performed additional analyses to investigate if there is any statistical significance for the asymmetry seen in the composite anomalies in the first row of Fig. 5. This is done by assessing the composites for the asymmetrical portion of the ENSO teleconnection (El Niño+La Niña) against the null hypothesis that they are statistically indistinguishable from zero. The results

are shown in Fig. 6. Firstly, in the top row, it is noted that there are no statistically significant values in the North Pacific and North Atlantic areas of interest in this study. In other words, we cannot reject the null hypothesis at the 5% significance level based on $t$ tests, meaning that we find no significant asymmetry in these regions. There are some locations at lower latitudes over the extratropical Pacific and Atlantic oceans which exhibit statistically significant nonlinearity. Secondly, the second row of Fig. 6 shows the Taylor diagrams for the asymmetrical portion of the teleconnecton over the North Atlantic. The confidence

intervals for the spatial correlations of the bootstrap composites and the observed composites have lower limits at -0.09 and -0.22 (blue lines), which are weak and even negative, indicating again that nonlinearity is not detectable, at least according to these domain-wise metrics. Note that these analyses are performed using observed SLP anomalies identified with Niño3.4 events as in other parts of this study. We do not investigate nonlinearity related to other factors such as ENSO SST types (except briefly below) or amplitudes, other atmospheric variables and regions, as this is a challenging question in its own right,

and is addressed by other studies (e.g., Garfinkel et al. 2013; Jiménez-Esteve and Domeisen 2020; Trascasa-Castro et al. 2019; Weinberger et al. 2019). Consistent with our results, these studies also generally agree that elucidation of nonlinearities using the limited sample of events in the reanalysis record is difficult, and therefore this research requires larger samples of model data.

       Many previous studies (e.g., Feng et al. 2017; Frauen et al. 2014; Garfinkel et al. 2013; Toniazzo and Scaife 2006; Zhang

et al. 2018) investigated asymmetry in ENSO teleconnections arising from central Pacific (CP) or eastern Pacific (EP) events, or due to varying ENSO strength. Zhang et al. (2018) show that CP-El Niño and CP-La Niña events are symmetrical (see Fig. A5e, g), while CP-La Niña and EP-La Niña (Fig. A5g, h) teleconnections in the Euro-Atlantic area are not the same either spatially or in sign (the CP– and EP–ENSO events selected are listed in Table 2). They postulate that the asymmetry can be due to the fact that as the EP-La Niña develops, the eastern tropical Pacific becomes progressively colder, thus reducing the chances

of overcoming the convection threshold to influence the overlying atmosphere and trigger teleconnections. They note, however, that they are unable to provide an explanation for the strong anomaly in the North Atlantic-Europe during EP-La Niña (Fig. A5h), which is not symmetrical with El Niño's. A brief check of the ND composites associated with these previously identified CP and EP events show very little in terms of robust teleconnection signals (Fig. A5a–d), especially over the North Atlantic. This is likely due to a combination of weaker ENSO SST anomalies in ND and smaller areas over which SST anomalies

occur during CP or EP events compared to all events pooled together. Studies such as those cited in the previous paragraph do suggest, however, that for JF, mixing CP- and EP-ENSO (as the Nino3.4 index does) may result in cancellation of signals in the North Atlantic, thus affecting the statistical significance and increasing the uncertainty of the signals. Therefore, asymmetry is perhaps a more important factor for JF than for ND.

### 3.3    Confidence intervals and $t$-test

Confidence intervals are important for quantifying uncertainty. Different types of confidence intervals have different accuracies depending on the properties of the data, therefore it is informative to consider more than one type (Hesterberg, 2015). The most common (without bootstrapping) is the "ordinary $t$ interval" ($ordt$) which is directly related to the $t$ test for statistical significance (see Sect. 2.3 and Table 3). The lower and upper limits (at 95% confidence) of this interval are shown in the first column of Fig. 7 for the SLP composite for JF El Niño (the full set for different events is presented in Figs. A6, A7).

Bootstrapping allows for confidence intervals that do not rely on assumptions about the sample distribution. The most intuitive type of confidence interval from bootstrapping is the "bootstrap percentile interval" ($perc$), where the percentiles are obtained directly from the values of the bootstrap composites in $\mathbb{C}^*$, and shown in the second column of Fig. 7 (also Figs. A6, A7). The confidence intervals used throughout Sect. 3.1 and by DES2017 are essentially of this type.

   Further types of confidence intervals which are not commonly considered in the climate research literature can be calculated
from bootstrapping. We describe two additional ones – $tBoot$ and $bootT$ (Table 3 and Hesterberg 2015). The $tBoot$ interval is also quite straightforward: the standard error ($\mathbf{SE}$ in $\mathbf{C}_o \pm t_{\alpha/2} \cdot \mathbf{SE}$) is calculated as the standard deviation of the bootstrap composites. This comes from the definition of standard error being the standard deviation of the sampling distribution, and using the bootstrap distribution as a substitute for the sampling distribution. For our SLP composites, these $tBoot$ intervals (not shown) are virtually identical to the $perc$ intervals. The $bootT$ interval is less immediately obvious but important because

it has higher accuracy (Hesterberg 2015) and, like $perc$, it allows for asymmetrical intervals about the mean. Instead of using a table, the $t$ values themselves are calculated from the bootstrap samples (of which we have 2000) and then the 2.5th and 97.5th percentiles of the $t$ values are obtained. To do this, from *each* sample we calculate a value of $\mathbf{t}^* = (\mathbf{C}^* - \overline{\mathbb{C}^*})/\mathbf{SE}^*$, where $\mathbf{C}^*$ is a bootstrap composite, $\overline{\mathbb{C}^*}$ indicates the mean of *all* the bootstrap composites, and $\mathbf{SE}^* = \mathbf{s}^*/\sqrt{n}$ is the standard error for this bootstrap sample. This is performed for all samples to obtain 2000 $t$ values. The percentiles of the $\mathbf{t}^*$ values are then used

to determine the $bootT$ confidence intervals (see equations in Table 3), and these are shown in the third column of Figs. 7, A6, A7.

   Comparing the $ordt$, $perc$ and $bootT$ confidence intervals, we note a few interesting aspects in Figs. A6, A7. Firstly, $ordt$ intervals are consistent (and must be by definition) with the absence or presence of statistically significant areas shown in Fig. 5.

For example, Fig. 7a, d show that $ordt$ in the North Pacific center and the southern center of the North Atlantic are bounded by negative values at the 95% confidence level, and the signals in these regions are indeed largely significant (Fig. 5). In contrast, the confidence interval for the northern center of the North Atlantic crosses "0" (includes negative and positive values), thus implying that the sign and amplitude of the signal are uncertain, consistent with the absence of statistical significance in Fig. 5c over this area. Secondly, all estimates of the confidence interval agree very well with each other, implying that the $ordt$ intervals are good enough for our purposes (although we would not know this *a priori*). Inspecting the different types of confidence intervals for these cases shows, however, that the $perc$ and $bootT$ intervals yield slightly larger areas of statistical significance in the nothern center North Atlantic teleconnections than $ordt$, suggesting that the bootstrap composites exhibit a small amount of skewness here.

Examining different types of confidence intervals, three of which are obtained from the bootstrap test, provides further support for the results shown in previous subsections. In particular, despite uncertainty in the amplitudes, we have reasonably high certainty in the signs of the teleconnection anomalies in the main centers of action shown by the ordinary $t$ test in Fig. 1. The bootstrap confidence intervals may also provide new information for regions where uncertainty in the ENSO response is not normally distributed, such as the northern center of the North Atlantic during late winter.

## 4   Concluding remarks

This study clarifies the uncertainty in the ENSO teleconnection in North Atlantic by considering the early (Nov-Dec) and late (Jan-Feb) parts of the cold season separately, as well as using a geographical area that better covers both anomaly centers in the North Atlantic. The motivation for separating the seasons in this way follows logically from previous studies that find different teleconnection signals and mechanisms at work during ND compared to JF. Various confidence intervals were used to assess uncertainty, including the widely used Student's $t$ test as well as several types of intervals calculated from the bootstrapping analysis (Sect. 3.3). These produce nearly the same results in most cases, thus indicating that the conventional Student's $t$ test (Fig. 1) and the equivalent $ordt$ confidence intervals (first columns of Figs. A6, A7) are generally reliable for assessing the statistical significance of ENSO teleconnection patterns.

The key results of this study are encapsulated in Figures 2 and 5. Based on our analyses of the observational record, we find that there is confidence in the spatial pattern of SLP anomalies associated with ENSO in the North Atlantic as it changes from November through February. There is an improved uncertainty than suggested by traditional DJF winter analyses, which average over opposite-signed anomalies in early and late winter. In agreement with DES2017, the analyses performed here indicate that uncertainty in the amplitudes of the teleconnection signals is indeed high, even when the cold-season transition is accounted for, with a 95th-to-5th percentiles ratio reaching a maximum factor of 2.8 in the North Atlantic (compared to a factor of 2.0 in the North Pacific). Overall, we argue that our analyses and conclusions permit an understanding of the ENSO teleconnection signals in the North Atlantic with reduced uncertainty compared to earlier studies which considered only the standard DJF winter season view.

After many decades of research, interest in ENSO teleconnections in the North Atlantic remains high and new results continue to appear (e.g., recent review by Domeisen et al. 2019). A number of recent studies have contributed to the continuous improvement in understanding the mechanisms behind the ENSO teleconnection to the North Atlantic, including details about how the late fall signal may arise from an atmospheric bridge via the tropical Atlantic (Ayarzagüena et al. 2018; King et al. 2018a) or diabatic heating over the Indian Ocean (Abid et al. 2021; Joshi et al. 2021). Such studies (and Molteni et al. 2020) have also found that "free-running" coupled model or SST-forced atmospheric model experiments do not, or only weakly, reproduce the ENSO teleconnection to the North Atlantic through the cold season, while intialized hindcasts perform better. Other studies have focused on the effects of the ENSO teleconnection on surface climate in Europe (precipitation, temperature, drought indices, e.g., Brönnimann et al. 2007; King et al. 2020; van Oldenborgh and Burgers 2005), for which model performance is less well documented than for atmospheric circulation anomalies (but see e.g., King et al. 2020; Volpi et al. 2020). Further research on surface climate impacts in models and prediction skills should consider the varying nature of the ENSO teleconnection through the cold season (discussion in King et al. 2018a).

*Code and data availability.* Supplementary code and data that may be used to reproduce the analyses and figures are archived at http://doi.org/10.5281/zenodo.4587405.

*Author contributions.* MPK conceived the initial concepts and performed the analyses. CL and SS provided suggestions for the analyses and figures. All authors contributed to the writing.

*Competing interests.* Camille Li is an Executive Editor of this journal.

*Acknowledgements.* This study is supported by Research Council of Norway (project nos. 255027, 275268). Stefan Sobolowski was partly supported by the EMULATE project, funded through basic institutional support from the Norwegian Department of Education to the Bjerknes Centre for Climate Research.

## Appendix A: Permutation test

In Sect. 3.1, the assessment of the composites is essentially performed by quantifying the uncertainty of the El Niño minus La Niña composites. Here, we demonstrate an alternative resampling technique called the permutation test. The null hypothesis is that the SLP anomalies under El Niño and La Niña events originate from the same population. First, the El Niño and La Niña years are put into the same pool. For example, for NDJ, the 26 El Niño and 22 La Niña years (see Table 1) are pooled together. Second, we randomly draw 26 years from the pool and reassign (also called relabel) them as El Niño, and the remaining 22

years as La Niña. The composite $\mathbf{C}^*$ is then calculated, and the steps are repeated to obtain 2000 composites as before. If the reassigned bootstrap composites are distinguishable from the original bootstrap composites, then the null hypothesis can be rejected, meaning that El Niño and La Niña events are different.

350    The resulting Taylor diagrams are shown in Fig. A4. For the spatial amplitude, we estimate a $p$ value for the SLP composite based on the proportion of bootstrap composites at least as large as the original composite: $30/2000 = 0.015$, $16/2000 = 0.008$, and $11/2000 \approx 0.005$ (the numerators of the fractions are the numbers of red dots in the panels, and the denominators are just the total number of bootstrap composites) for DJF, ND, and JF, respectively. Another way to describe these $p$ values is that each one is the probability of obtaining an SLP composite at least as large as the observed one if El Niño and La Niña related

355    SLP anomalies were from the same population. These $p$ values are small enough (following common practice of requiring for e.g. $p = 0.05$) to reject the null hypothesis. Note also that none of the bootstrap confidence intervals for the amplitude ratios, indicated with the blue and red semi-circles, crosses over the gray semi-circles representing the observed amplitudes (therefore radii for red semi-circles $< 1$). The permutation tests carried out here suggest that the SLP anomalies associated with El Niño and La Niña in the North Atlantic can be distinguished from each other with a high confidence. This is equivalent and consistent

360    to the result in Sect. 3.1.

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

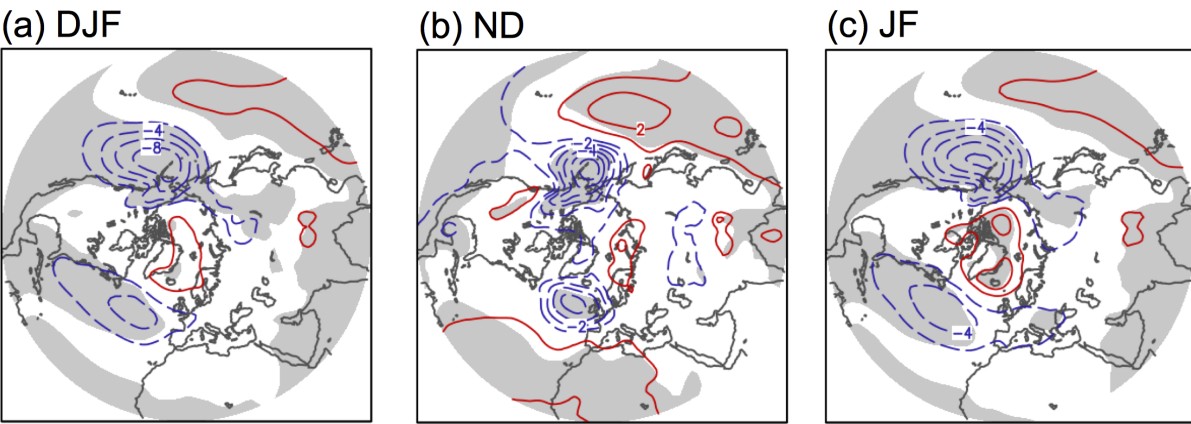

**Figure 1.** Sea level pressure (SLP) composites for El Niño minus La Niña events during 1920–2014 for the months shown on the panels. The ENSO events are defined by Nino3.4 amplitudes greater than 1 standard deviation. Contour interval is 2 hPa in (a) and (c) and 1 hPa in (b), with red (blue) contours indicating positive (negative) anomalies. Gray shading indicates the 5% significance level for a two-tailed $t$ test. All analyses in this paper use data from HadSST and NOAA-CIRES 20CR V2c.

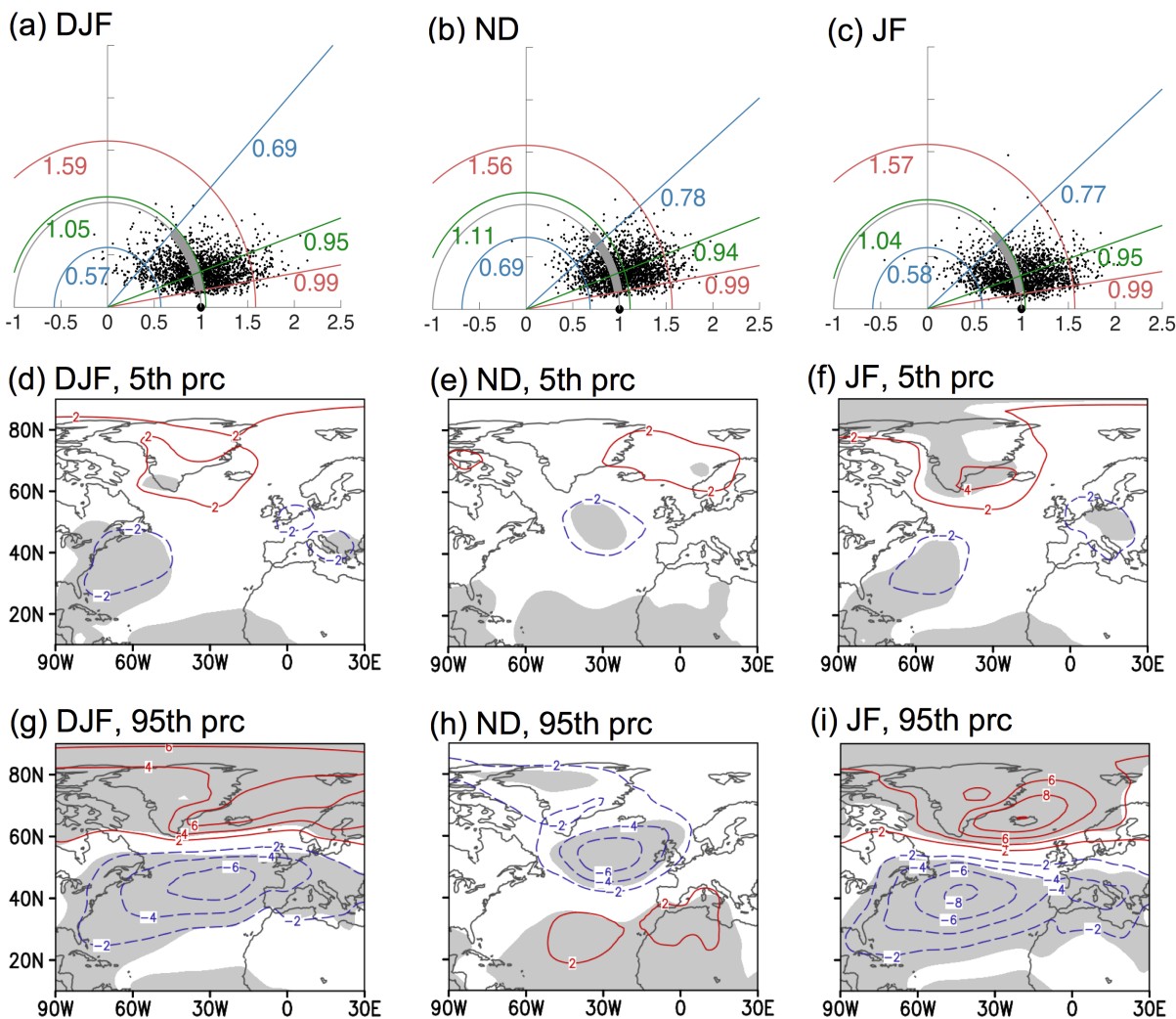

**Figure 2.** Uncertainty of ENSO (El Niño minus La Niña) teleconnection in the North Atlantic during 1920-2014 for the months shown. Top row: Taylor diagrams for North Atlantic with 2000 bootstrap SLP composites (represented by the small dots). The blue, green, and red semi-circles indicate the 5th, median, and 95th percentiles of the amplitude ratios. Blue, green, and red lines indicate the 5th, median, and 95th percentiles of the spatial correlation with the observed composite (represented by the large dot). Second and third rows: Bootstrap SLP composites corresponding to the 5th or 95th percentile based on both amplitude ratio and spatial correlation. Contour interval is 2 hPa, with red (blue) contours indicating positive (negative) anomalies. Gray shading indicates the 5% significance level for a two-tailed $t$ test. See Sects. 2.4, 3.1 for more details.

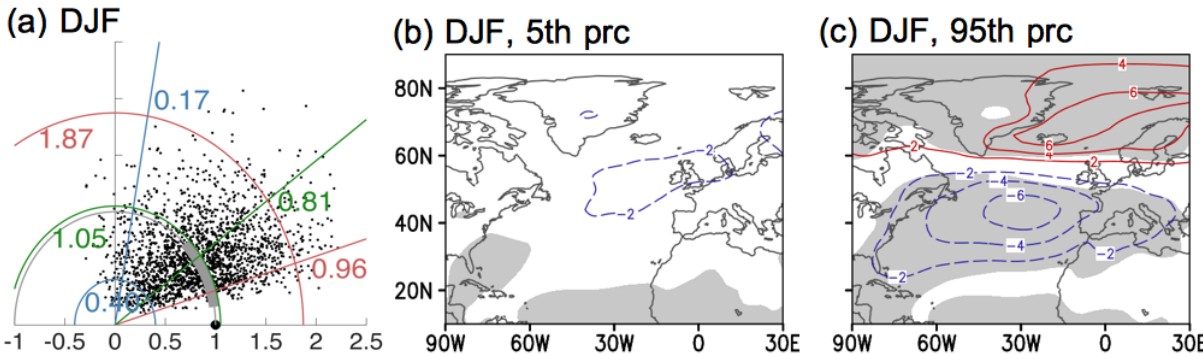

**Figure 3.** Taylor diagram for polar (poleward of $60°$N) North Atlantic in DJF with 2000 bootstrap SLP composites, and the 5th and 95th percentile maps. These panels correspond to the first column in Fig. 2. Features in the Taylor diagram are as described in Fig. 2. See Sect. 3.1 for more details.

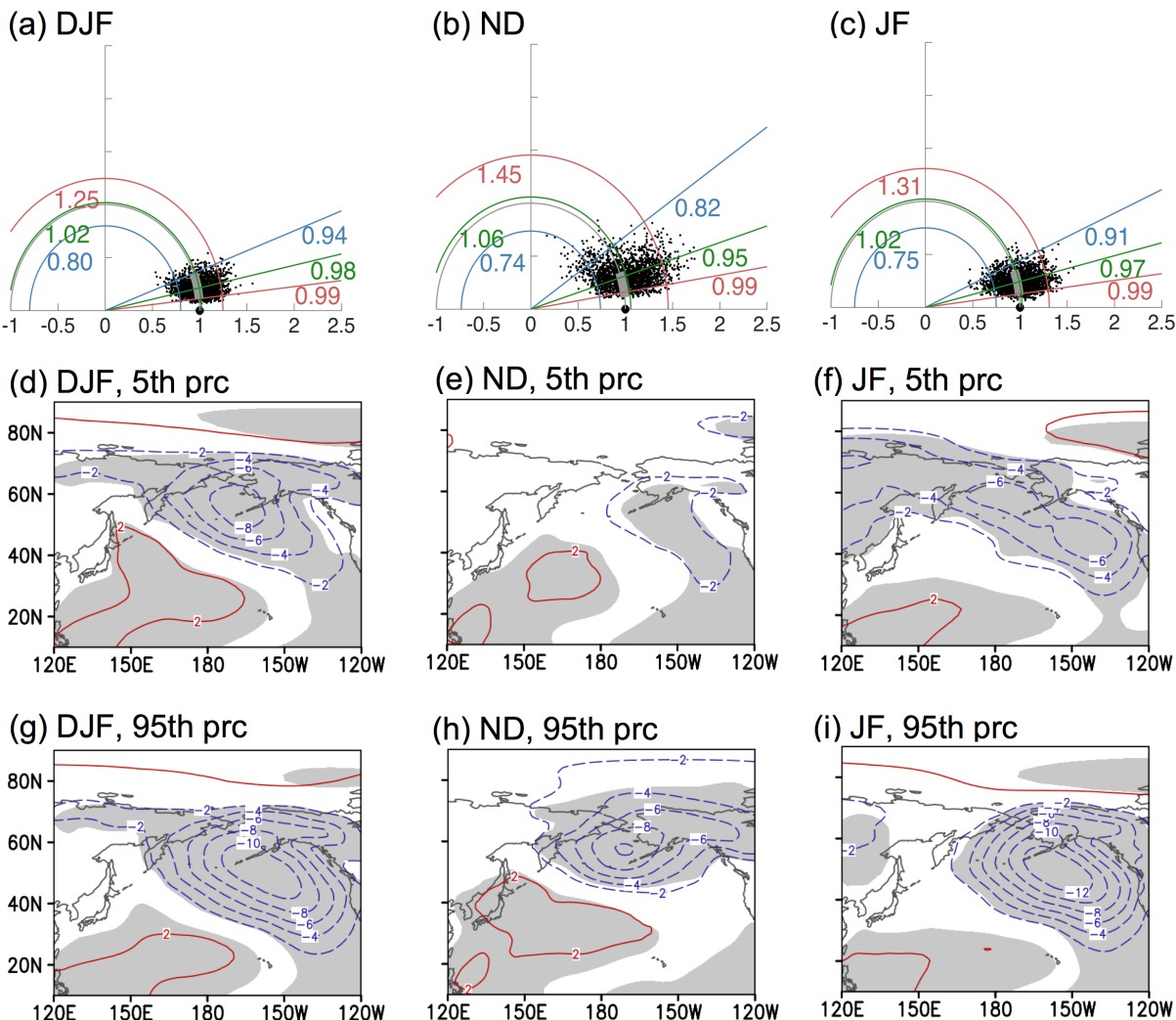

**Figure 4.** Same as Fig. 2 except for the North Pacific.

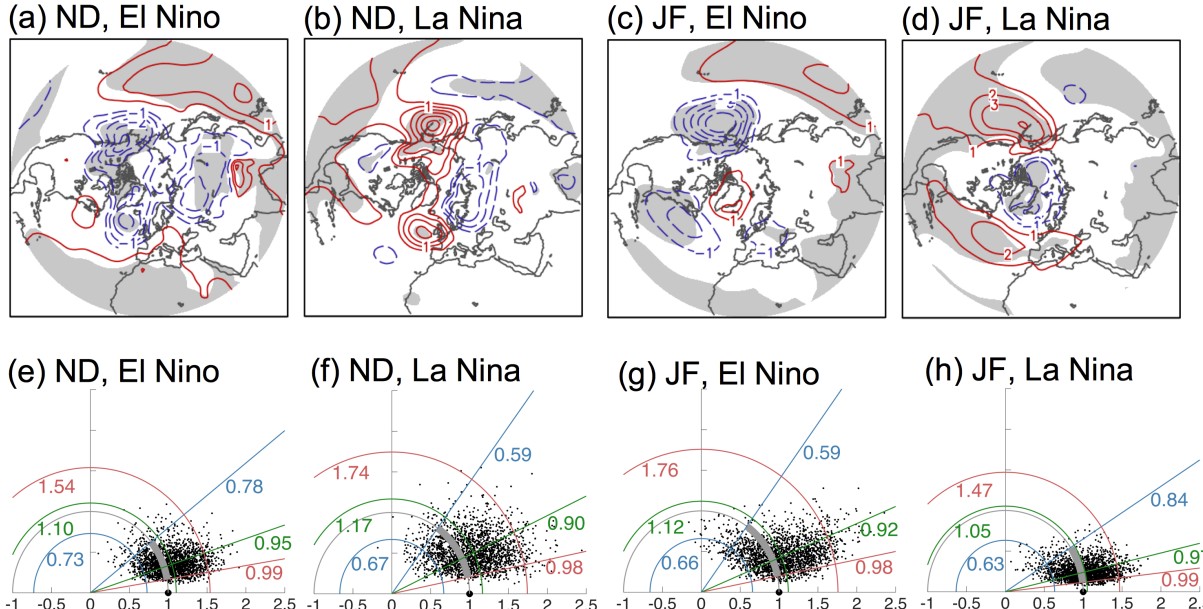

**Figure 5.** Uncertainty of ENSO (for El Niño or La Niña separately) teleconnection in the North Atlantic during 1870-2014 for the months shown. Top row: SLP composites for El Niño or La Niña events. Contour interval is 0.5 hPa for ND and 1 hPa for JF, with red (blue) contours indicating positive (negative) anomalies. Gray shading indicates 5% significance level for a two-tailed $t$ test. Second row: Corresponding Taylor diagrams for North Atlantic with 2000 bootstrap SLP composites. Features in the Taylor diagrams are as described in Fig. 2. See Sects. 2.4, 3.2 for more details.

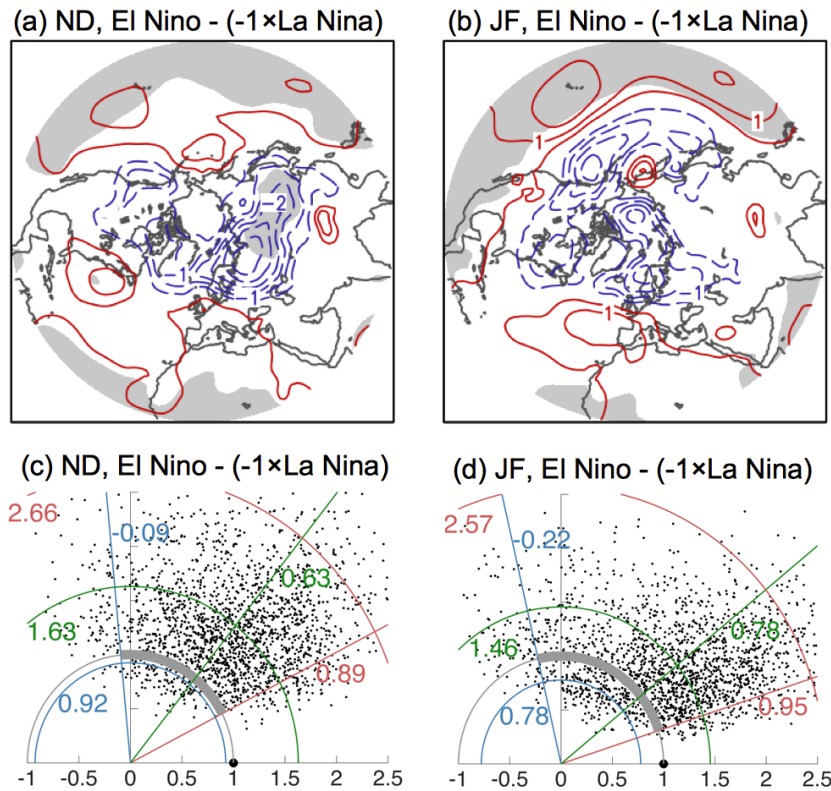

**Figure 6.** Top row: Asymmetrical portion (El Niño-(-1×La Niña)) of the ENSO teleconnection composites in Fig. 5. Red (blue) contours indicate positive (negative) anomalies, with gray shading indicating the 5% significance level for a two-tailed $t$ test (the null hypothesis is the difference of El Niño and -1×La Niña is statistically indistinguishable from zero). Contour interval is 0.5 hPa. Second row: Taylor diagrams for North Atlantic with 2000 bootstrap composites of El Niño - (-1×La Niña), as shown in a, b. Features in the Taylor diagram are as described in Fig. 2. See Sect. 3.2 for more details.

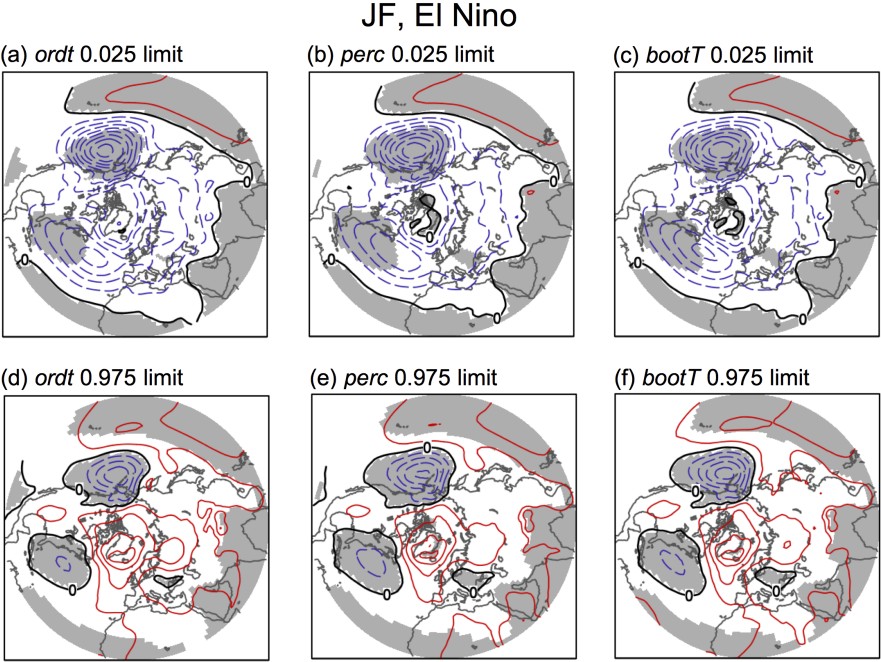

**Figure 7.** Lower and upper limits of confidence intervals (at 95% level) for JF El Niño SLP composite corresponding to Fig. 5c. The *ordt*, *perc*, and *bootT* intervals are described in Sect. 3.3. Contour interval is 1 hPa, with red (blue) contours indicating positive (negative) anomalies. Gray shading indicates locations where the SLP anomalies have the same sign within the entire interval.

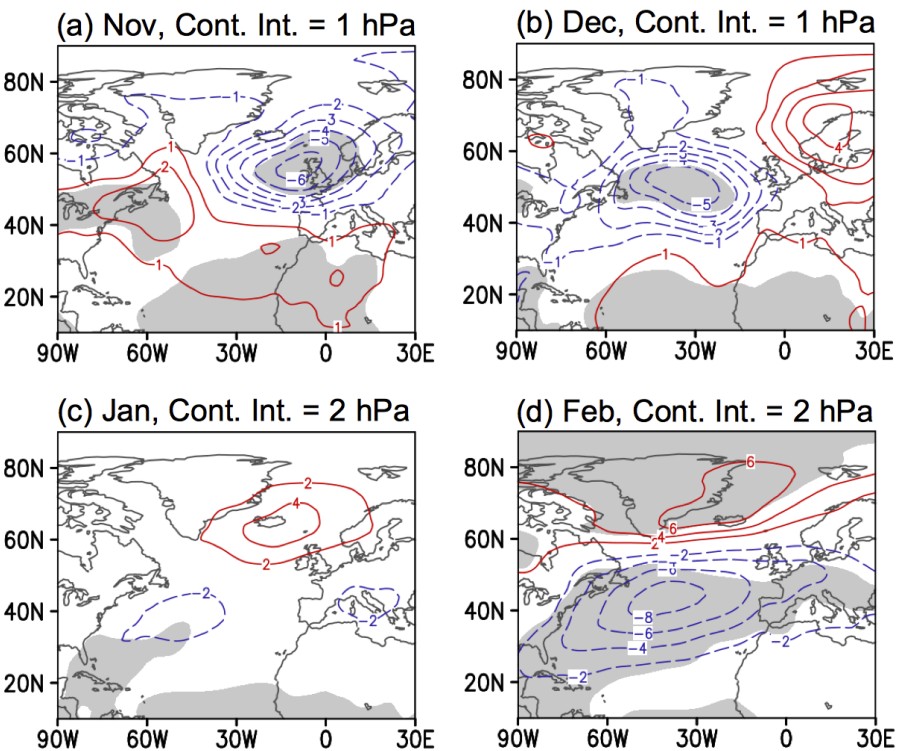

**Figure A1.** SLP composites for El Niño minus La Niña events during 1920–2014 for the months shown on the panels. These are similar to Fig. 1 except here composites for the 4 months are calculated individually. Red (blue) contours indicating positive (negative) anomalies, with gray shading indicates the 5% significance level for a two-tailed t test.

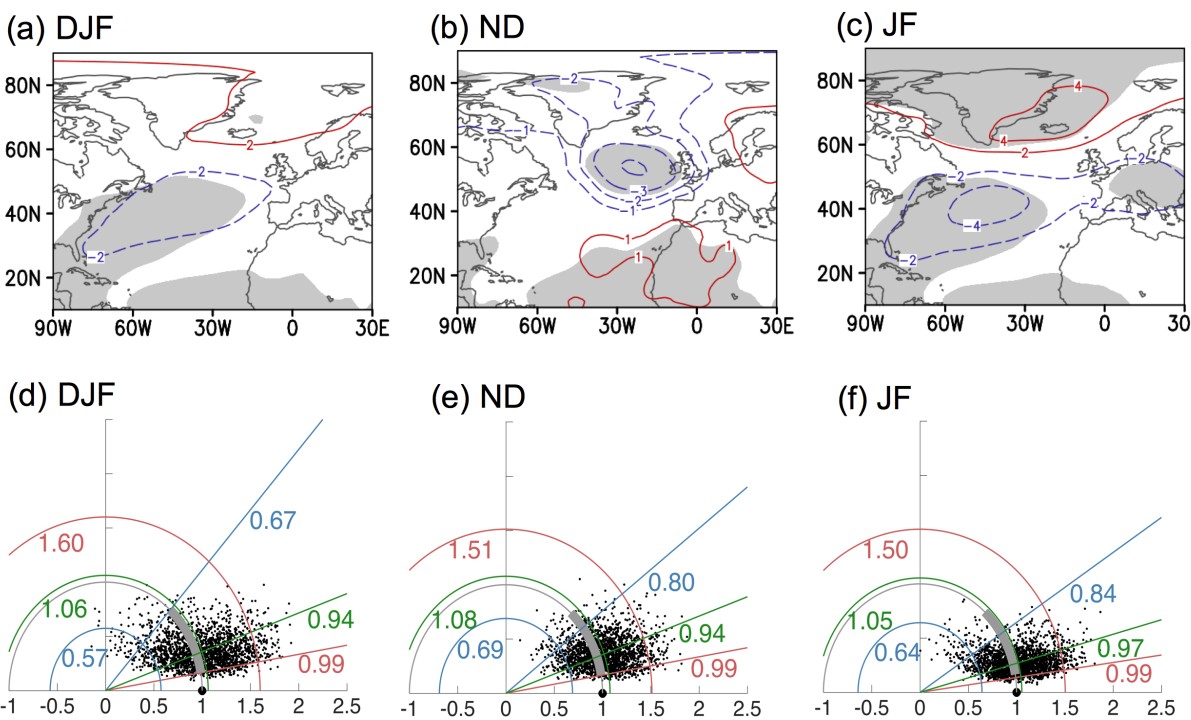

**Figure A2.** Similar to Figs. 1, 2 except for a longer period of 1870–2014. Top row: SLP composites for El Niño minus La Niña events. Contour interval in (a) and (c) is 2 hPa, and is 1 hPa in (b), with red (blue) contours indicating positive (negative) anomalies. Gray shading indicates 5% significance level for a two-tailed $t$ test. Second row: Corresponding Taylor diagrams for North Atlantic with 2000 bootstrap SLP composites. Features in the Taylor diagrams are as described in Fig. 2. See Sects. 2.4, 3.1 for more details.

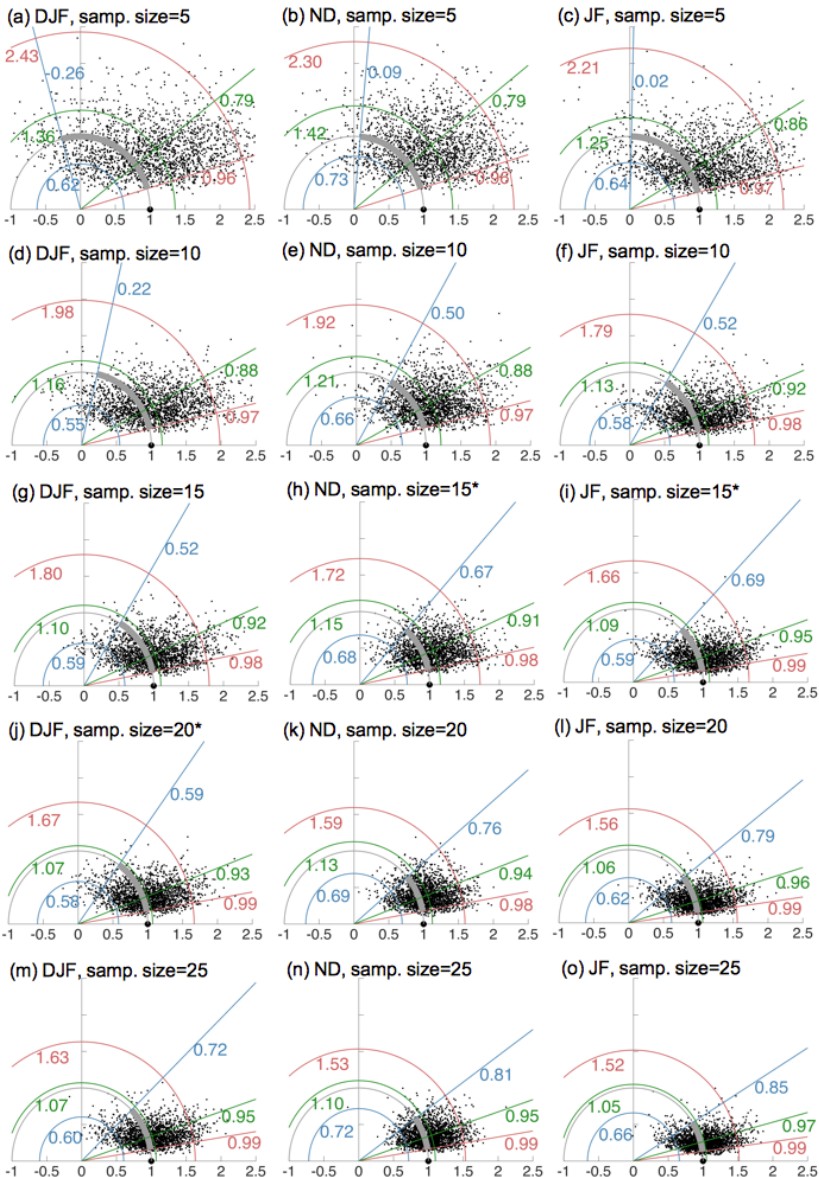

**Figure A3.** Each panel shows a Taylor diagram for North Atlantic with 2000 bootstrap SLP composites, of which each has a sample size shown in the panel title. Each sample is bootstrapped (sub-sampling with replacements) from events in 1870-2014 (Table 1). These panels correspond to the second row of Fig. A2 except for different sample sizes. Panels j, h, i indicate the estimated minimum sample sizes required for robust signals for spatial patterns of the composites. See Sect. 3.1 for more details.

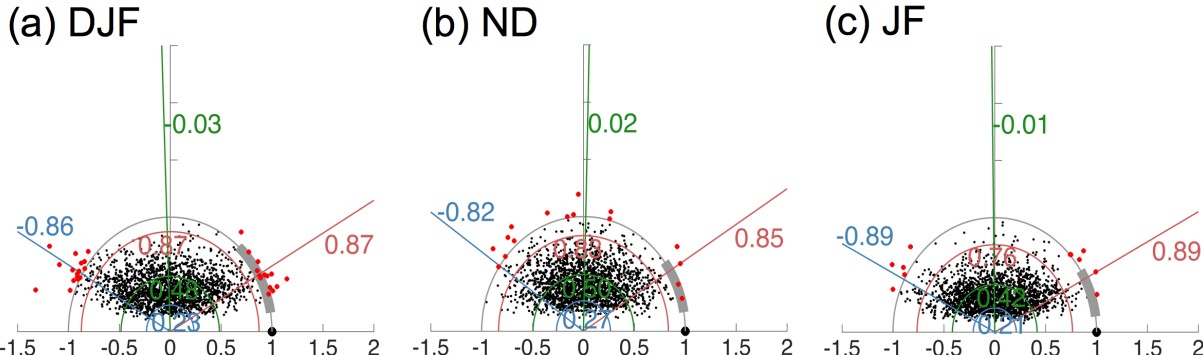

**Figure A4.** Taylor diagrams for permutation tests corresponding to the same events in Fig. A2 (1870-2014). The thick gray arc marks the 5th-to-95th percentile range in the corresponding panel in Fig. A2. Other features in the Taylor diagrams are as described in Fig. 2. See Appendix A for more details.

Nov/Dec

Jan/Feb

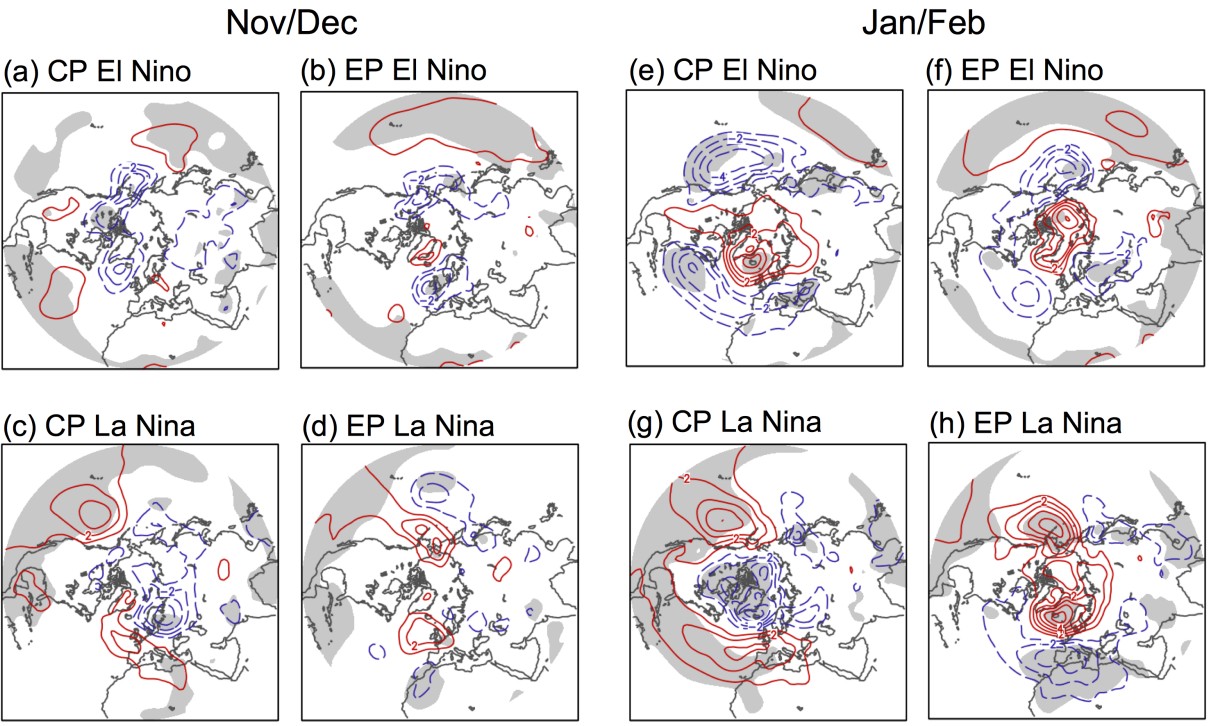

(a) CP El Nino  (b) EP El Nino  (e) CP El Nino  (f) EP El Nino

(c) CP La Nina  (d) EP La Nina  (g) CP La Nina  (h) EP La Nina

**Figure A5.** SLP composites for central- and eastern-Pacific El Niño, and La Niña, events in the ND or JF means. Contour interval is 1 hPa in all panels, with red (blue) contours indicating positive (negative) anomalies. Gray shading indicates 5% significance level for a two-tailed $t$ test. See Sect. 3.2 for more details.

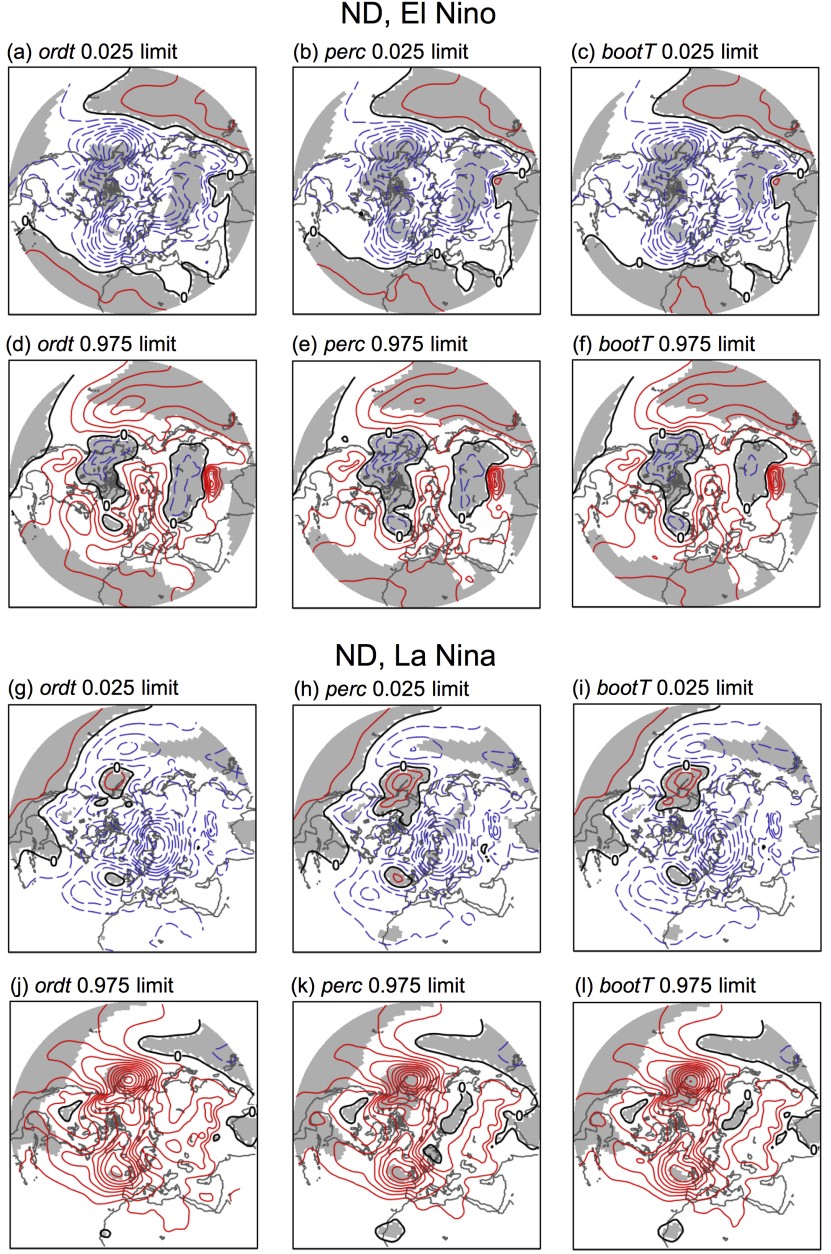

**Figure A6.** Lower and upper limits of confidence intervals (at 95% level) for ND El Niño and La Niña SLP composites corresponding to Fig. 5a, b. The $ordt$, $perc$, and $bootT$ intervals are described in Sect. 3.3. Contour interval is 0.5 hPa, with red (blue) contours indicating positive (negative) anomalies. Gray shading indicates locations where the SLP anomalies have the same sign within the entire interval.

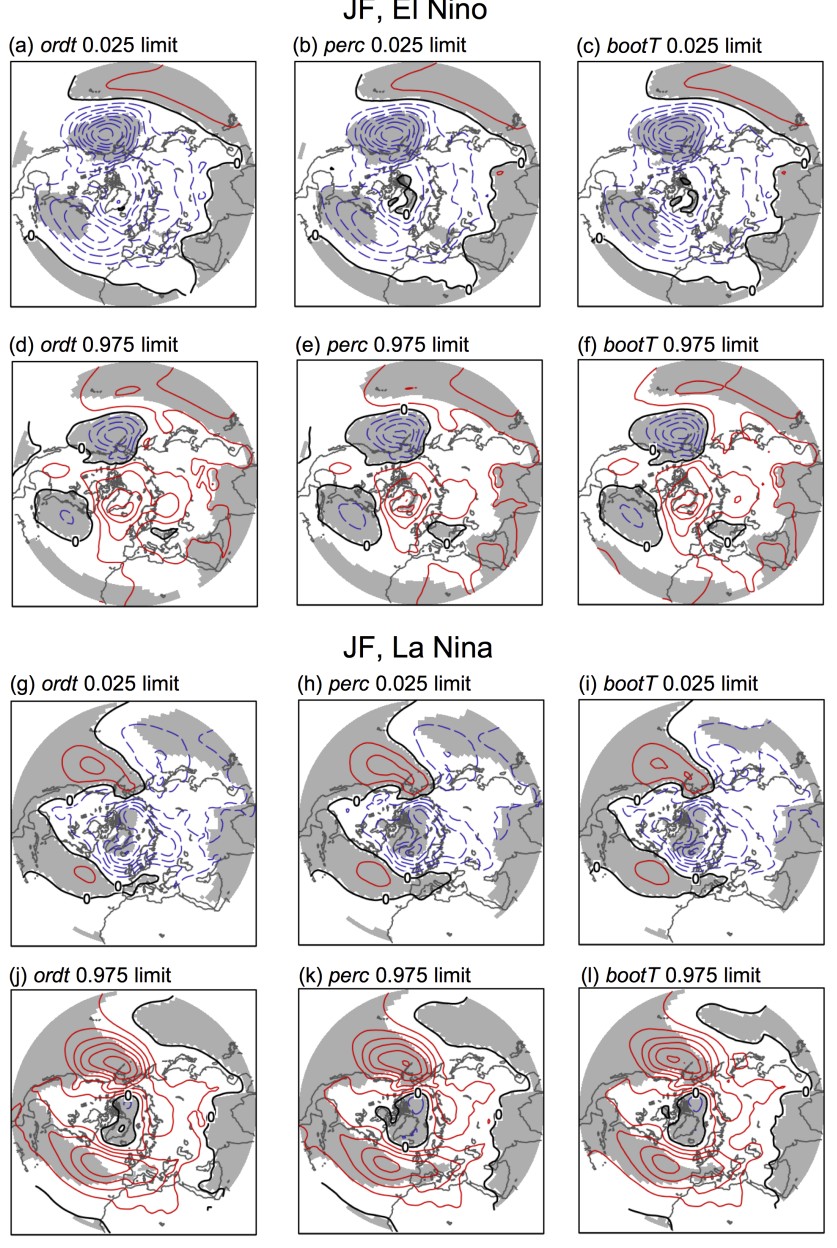

**Figure A7.** Lower and upper limits of confidence intervals (at 95% level) for JF El Niño and La Niña SLP composites corresponding to Fig. 5c, d. The *ordt*, *perc*, and *bootT* intervals are described in Sect. 3.3. Contour interval is 1 hPa, with red (blue) contours indicating positive (negative) anomalies. Gray shading indicates locations where the SLP anomalies have the same sign within the entire interval.

**Table 1.** ENSO events based on Niño3.4 used in the composite calculations.

**Type (number of events):** Years

**El Niño, NDJ or DJ, 1920–2014 (18):** 1925/26, 1930/31, 1940/41, 1941/42, 1957/58, 1963/64, 1965/66, 1972/73, 1977/78, 1982/83, 1986/87, 1987/88, 1991/92, 1994/95, 1997/98, 2002/03, 2006/07, 2009/10.

**La Niña, NDJ or DJ, 1920–2014 (16):** 1924/25, 1933/34, 1942/43, 1949/50, 1950/51, 1955/56, 1970/71, 1973/74, 1975/76, 1984/85, 1988/89, 1998/99, 1999/00, 2007/08, 2010/11, 2011/12.

**El Niño, ON, 1920–2014 (18):** 1923, 1925, 1930, 1941, 1957, 1963, 1965, 1972, 1977, 1982, 1986, 1987, 1991, 1994, 1997, 2002, 2006, 2009.

**La Niña, ON, 1920–2014 (17):** 1933, 1942, 1949, 1950, 1954, 1955, 1964, 1973, 1975, 1983, 1984, 1988, 1998, 1999, 2007, 2010, 2011.

**El Niño, NDJ or DJ, 1870–2014 (26):** 1877/78, 1888/89, 1896/97, 1899/00, 1902/03, 1905/06, 1911/12, 1918/19, in addition to those in 1920–2014 given above.

**La Niña, NDJ or DJ, 1870–2014 (22):** 1886/87, 1889/90, 1892/93, 1893/94, 1909/10, 1916/17, in addition to those in 1920–2014 given above.

**El Niño, ON, 1870–2014 (25):** 1877, 1885, 1888, 1896, 1899, 1902, 1905, 1918, in addition to those in 1920–2014 given above except 2006.

**La Niña, ON, 1870–2014 (23):** 1874,1886,1889,1892,1893,1909,1916, in addition to those in 1920–2014 given above except 1954.

**Table 2.** Central–Pacific (CP) and Eastern–Pacific (EP) ENSO events (following Zhang et al. 2018) used for calculating the composites in Fig. A5. See also Sect. 3.2.

| Type (number of events): Years |
| --- |
| **CP El Niño (9):** 1953/54, 1957/58, 1968/69, 1977/78, 1979/80, 1986/87, 2002/03, 2004/05, 2009/10. |
| **EP El Niño (8):** 1951/52, 1952/53, 1963/64, 1965/66, 1969/70, 1972/73, 1976/77, 1997/98. |
| **CP La Niña (8):** 1973/74, 1974/75, 1975/76, 1988/89, 1998/99, 2000/01, 2010/11, 2011/12. |
| **EP La Niña (8):** 1954/55, 1955/56, 1964/65, 1967/68, 1971/72, 1984/85, 1995/96, 2005/06. |

**Table 3.** Types of confidence intervals described Sect. 3.3 and Hesterberg (2015). The subscript $\alpha/2$ indicates a two-tailed $t$ value at $\alpha$ significance level. e.g. The current studies uses $\alpha = 0.05$.

| Long name | Short name | General equation/descriptions |
|---|---|---|
| Ordinary $t$ interval | $ordt$ | $\mu = \mathbf{C}_o \pm t_{\alpha/2} \cdot \mathbf{SE}_o$, where $\mathbf{SE}_o$ is estimated from original sample |
| Bootstrap percentile (or confidence) interval | $perc$ | Intervals are obtained from bootstrap composites directly |
| $t$ interval with bootstrap SE | $tBoot$ | $\mathbf{SE}$ is estimated as standard deviation of bootstrap composites |
| Bootstrap $t$ interval | $bootT$ | $\mu = \mathbf{C}_o - bootT_{1-\alpha/2} \cdot \mathbf{SE}_o$, $\mu = \mathbf{C}_o - bootT_{\alpha/2} \cdot \mathbf{SE}_o$, where $\mathbf{SE}_o$ is estimated from original sample, and $bootT$ values are calculated from all bootstrap samples |