# Peer review of "Resampling of ENSO teleconnections: accounting for cold season evolution reduces uncertainty in the North Atlantic"

_Weather and Climate Dynamics, 2021_

## Author Comment (AC1)

Thank you for the review of our manuscript. In this final authors' comments, we give a few immediate/early replies to selected comments. All comments will be given full attention and point-by-point replies will be provided with the revised manuscript later.

```
Reviewer 1.
```

**General comments:**

1. Given that the authors argue that splitting ND from JF is important, I think it is important to show each of these 4 months individually in a version of figure 2 with four columns and three rows, before then showing the ND and JF composites. This additional figure would help make the point to the reader that ND are more alike, and JF are more alike, before the authors then go on to combine ND together and JF together.

We show below the regressions (not composites) of SLP in Oct, Nov, Dec, and Jan separately on HadISST Nino3.4 plotted quickly using the KNMI Climate Explorer web tool (https://climexp.knmi.nl/start.cgi).

[Figure]

It is seen here that there is an evolution through these months. In particular, the signs in the two major centres of action change from November/December to January/February.

In the revision, we will carry out additional analysis for these months using the method consistent with the rest of the manuscript and document the result appropriately.

2. There is one recent paper the authors seem to have missed that is very relevant and complementary to the present study. The size of a composite necessary for the stratospheric route in late winter to become robust was discussed in detail in Weinberger et al 2019. They concluded that the EN-weak vortex route is robust with around 15 events, though the impact of LN on the vortex was only apparent with more than 25 events (see their figure 5) as compared to climatology. They don't show the difference between EN and LN, but clearly the difference will be significant with less.

The polar cap SLP response was also found to be significant after around 15 events are taken (their figure 7) both for EN and LN. This is consistent with the results shown in the present paper.

Weinberger et al also found robust impacts on subpolar Eurasian temperatures (see their figure 7) that are robust for EN and LN individually with composite sizes similar to that used in the present paper. Weinberger et al didn't consider early winter and they analyzed historical data for a much shorter period than analyzed here, and so there are clearly new results here. But as best as I can tell the results are consistent when the late winter and the stratospheric route is considered.

Weinberger, I., Garfinkel, C.I., White, I.P. and Oman, L.D., 2019. The salience of nonlinearities in the boreal winter response to ENSO: Arctic stratosphere and Europe. Climate dynamics, 53(7), pp.4591-4610.

Thank you for bringing our attention to this study. We will add discussion related to it. The question of statistical significance and its dependence on sample size (i.e., number of events) is an interesting one. We have not studied nor reported this thoroughly in our submitted manuscript (although we have done some exploratory tests previously). As reported in the manuscript, increasing the sample size to include events in the 19th century, compared to the post-1920 events used by Deser et al. 2017 (see Table 1), does not change the findings.

As we are using reanalysis data, we have much smaller number of original (the complete non-resampled) events compared to Weinberger et al. with their analysis on a model ensemble. However, bootstrapping does permit us to re-sample the available events to smaller sample sizes (as well as to larger ones if certain assumptions are made), and therefore the effect of sample sizes in observation/reanalysis can also be estimated.

We are interested to investigate this more thoroughly and will decide on an appropriate method to document the result without changing the focus or essence of the current manuscript.

---

## Author Comment (AC2)

Thank you for the review of our manuscript. In this final authors' comments, we give a few immediate/early replies to selected comments. All comments will be given full attention and point-by-point replies will be provided with the revised manuscript.

Reviewer 2.

**General comments.**

1. The authors seem to have missed a few recent and relevant studies examining the asymmetry and non-linearity in the ENSO teleconnection to the North Atlantic. It would be nice that the papers cited below are included in the new version of the manuscript. At least when discussing that these model results disagree in some points when trying to identify nonlinearities in the ENSO-North Atlantic teleconnection (e.g. lines 194-201)

Hardiman, S. C., Dunstone, N. J., Scaife, A. A., Smith, D. M., Ineson, S., Lim, J., & Fereday, D. (2019). The Impact of Strong El Niño and La Niña Events on the North Atlantic. *Geophysical Research Letters*, *46*(5), 2874–2883. https://doi.org/10.1029/2018GL081776

Jiménez-Esteve, B., & Domeisen, D. I. V. (2020). Nonlinearity in the tropospheric pathway of ENSO to the North Atlantic. *Weather and Climate Dynamics*, *1*(1), 225–245. https://doi.org/10.5194/wcd-1-225-2020

Trascasa-Castro, P., Maycock, A. C., Scott Yiu, Y. Y., & Fletcher, J. K. (2019). On the Linearity of the Stratospheric and Euro-Atlantic Sector Response to ENSO. *Journal of Climate*, *32*(19), 6607–6626. https://doi.org/10.1175/JCLI-D-18-0746.1

Weinberger, I., Garfinkel, C. I., White, I. P., & Oman, L. D. (2019). The salience of nonlinearities in the boreal winter response to ENSO: Arctic stratosphere and Europe. *Climate Dynamics*, *53*(7-8), 4591–4610. https://doi.org/10.1007/s00382-019-04805-1

Thank you for the additional references. We will add discussion of these papers in the appropriate place in the revised manuscript. While our focus was not on asymmetry or non-linearity in ENSO teleconnection, this is of course an important issue. We report only that we have not found asymmetry in terms of the signs of the SLP anomaly with analyses based on Nino3.4. This finding agrees with that of Deser et al (2017) which used the same general approach as ours. In a future study, we plan to perform further hypothesis tests on nonlinearities in amplitudes of the teleconnection as well as anomalies related to different ENSO types. This could be challenging because non-linearities in the signals could be smaller (because one is taking differences of differences/anomalies), and stratifying the ENSO types using reanalysis/observation reduces the sample sizes.

We would like to keep the focus and key message for current manuscript. Therefore, while we will include the additional discussion, we do not plan to address the more complex issue of nonlinearities and effects of ENSO types.

2. I agree with reviewer 1, that it would be good to show the individual monthly mean evolution of the teleconnection from Nov to March.

(Same reply is also given to Reviewer 1) We show below the regressions (not composites) of SLP in Oct, Nov, Dec, and Jan separately on HadISST Nino3.4 plotted quickly using the KNMI Climate Explorer web tool (https://climexp.knmi.nl/start.cgi).

[Figure]

It is seen here that there is an evolution through these months. In particular, the signs in the two major centres of action change from November/December to January/February.

In the revision, we will carry out additional analysis for these months using the method consistent with the rest of the manuscript and document the result appropriately.

3. The authors analyze the effect of separating the ENSO events into CP and EP events. I think, given the nature of the paper, it would be important to analyze what is the effect of the ENSO magnitude, i.e. consider strong (Nino3.4>1.5/2SD) and moderate (1SD>Nino3.4>1.5/2SD) ENSO events separately. I think the current length of the paper would allow for the addition of this analysis.

We can carry out the same analysis and test based on this separation you suggest. It will be interesting to see how the previous finding on extreme ENSO teleconnection stand under the same bootstrap tests. For the case of strong/extreme events, we have some concerns on the potentially much smaller sample, and therefore the robustness of the conclusion that can be made. However, we will carry out the analyses and report the results in the revised manuscript.

4. Section 3.3 discusses different methods to compute confidence intervals. However, the conclusion of this section is rather "boring". Figures 7 and 8 look almost the same for all the different methods employed. I understand that the reason why the authors have decided to put these figures in the main text is to show that they are actually very similar. Nevertheless, I would recommend

```
putting these two figures as an appendix as they do not provide a lot of
quantitative information, but they are a justification for the methods employed
in the first part of the study.
```

Indeed, we carried out the analyses/tests in sect 3.3 to check what these different methods produce. The result is somewhat "boring" in the end, but we did not know this beforehand. In fact, the "boring-ness" could be considered the important part of the result. The additional tests are not normally employed by the wider climate teleconnection research community, and they might produce different results for other regions and teleconnection drivers.

However, we understand your point about the figures. We think it could work to use only selected panels in the main text body and move the full sets of panels to the appendix.

---

## Author Response (AR1)

We thank both reviewers for their comments. Here we reply to and address all of your comments (responses are in blue, Times New Roman font). The corresponding places where changes are made in the revised manuscript (tracked changes version) are tagged accordingly. Following the guideline, the tracked changes version of the revised manuscript is appended. A 'clean' version of the revised manuscript is also uploaded.

**Reviewer 1.**

**General comments:**

**1.** Given that the authors argue that splitting ND from JF is important, I think it is important to show each of these 4 months individually in a version of figure 2 with four columns and three rows, before then showing the ND and JF composites. This additional figure would help make the point to the reader that ND are more alike, and JF are more alike, before the authors then go on to combine ND together and JF together.

We have now added further explanation in the second paragraph of Sect. 3.1 as well as a new Fig. A1. Details of the teleconnection in the four months individually have also been presented in detail by previous studies (e.g. in the cited Moron and Gouirand 2003, among others). **Tag: R1.1, R2.2**

**2.** There is one recent paper the authors seem to have missed that is very relevant and complementary to the present study. The size of a composite necessary for the stratospheric route in late winter to become robust was discussed in detail in Weinberger et al 2019. They concluded that the EN-weak vortex route is robust with around 15 events, though the impact of LN on the vortex was only apparent with more than 25 events (see their figure 5) as compared to climatology. They don't show the difference between EN and LN, but clearly the difference will be significant with less.

The polar cap SLP response was also found to be significant after around 15 events are taken (their figure 7) both for EN and LN. This is consistent with the results shown in the present paper.

Weinberger et al also found robust impacts on subpolar Eurasian temperatures (see their figure 7) that are robust for EN and LN individually with composite sizes similar to that used in the present paper. Weinberger et al didn't consider early winter and they analyzed historical data for a much shorter period than analyzed here, and so there are clearly new results here. But as best as I can tell the results are consistent when the late winter and the stratospheric route is considered.

Thank you for pointing out this paper and summarising the relevant parts for us. We find that this is an interesting and important aspect. Therefore, although you have not requested it explicitly, we have performed additional calculations to investigate the effect of sample sizes in our cases. The result is described in a new paragraph in Sect 3.1 and a new appendix figure. Weinberger et al.'s work is also cited now. We hope you will find the outcome interesting and agree with our revision. **Tag: R1.2**

**Minor changes**

**Line 15 :** I suggest adding a more general first paragraph about ENSO teleconnections. The paper jumps right in to tackling the uncertainty, and while most of the intended audience probably already knows about ENSO teleconnections, a more casual reader may not.

We have now added a general background paragraph at the beginning of the Introduction. **Tag: R1.3**

**Figure 2 and similar:** Please add to the caption and to the text the region over which the spatial correlation is computed for the Taylor diagram. Is it identical to the region shown in the bottom rows? Is area-weighting applied?

This information is now added in Sect. 2.4 and the figure captions also refer to this section. Yes, both the spatial correlation and the spatial amplitude are area-weighted. This is also mentioned and clarified in Sect 2.4. **Tag: R1.4**

**Line 195 and table 1:** please clarify how you classify years as either CP-EN, EP-EN, EP-LN, or CP-LN. It would also be appreciated if a table was added that lists which years are included in each of the four composites (or alternately mark in bold or some other designation either EP or CP events on table 1).

A new Table 2 listing the years is added and also referred to in Sect. 3.2. **Tag: R1.5**

The years/events selected follow the same ones in the cited Zhang et al. (2018). We realise that there are variations in defining CP and EP ENSO and didn't want to get into investigating the subtlety deeply. We only wanted to check briefly what the Nov-Dec composites look like under these events selected by a previous study.

**Line 306 the publication year for the Domeisen et al review paper should be changed to 2019**

Corrected. Please see References. **R1.6**

**Reviewer 2.**

**General comments.**

**1.** The authors seem to have missed a few recent and relevant studies examining the asymmetry and non-linearity in the ENSO teleconnection to the North Atlantic. It would be nice that the papers cited below are included in the new version of the manuscript. At least when discussing that these model results disagree in some points when trying to identify nonlinearities in the ENSO-North Atlantic teleconnection (e.g. lines 194-201)

Hardiman, S. C., Dunstone, N. J., Scaife, A. A., Smith, D. M., Ineson, S., Lim, J., & Fereday, D. (2019). The Impact of Strong El Niño and La Niña Events on the North Atlantic. *Geophysical Research Letters*, *46*(5), 2874–2883. https://doi.org/10.1029/2018GL081776

Jiménez-Esteve, B., & Domeisen, D. I. V. (2020). Nonlinearity in the tropospheric pathway of ENSO to the North Atlantic. *Weather and Climate Dynamics*, *1*(1), 225–245. https://doi.org/10.5194/wcd-1-225-2020

Trascasa-Castro, P., Maycock, A. C., Scott Yiu, Y. Y., & Fletcher, J. K. (2019). On the Linearity of the Stratospheric and Euro-Atlantic Sector Response to ENSO. *Journal of Climate*, *32*(19), 6607–6626. https://doi.org/10.1175/JCLI-D-18-0746.1

Weinberger, I., Garfinkel, C. I., White, I. P., & Oman, L. D. (2019). The salience of nonlinearities in the boreal winter response to ENSO: Arctic stratosphere and Europe. *Climate Dynamics*, *53*(7–8), 4591–4610. https://doi.org/10.1007/s00382-019-04805-1

Thank you for highlighting the relevance of these papers. The last three papers are cited now in Sect. 3.2. **Tag: R2.1**

Please also see replies to your related comments R2.25, 2.26 below.

Hardiman et al. is cited in the newly added first paragraph in the Introduction (also see reply to R1.3). Weinberger et al. is also cited in Sect 3.1 (also reply to R1.2).

**2.** I agree with reviewer 1, that it would be good to show the individual monthly mean evolution of the teleconnection from Nov to March.

(Same reply to Reviewer 1) We have now added further explanation in the second paragraph of Sect. 3.1 as well as a new Fig. A1. Details of the teleconnection in the four months individually have also been presented in previous studies (e.g. in the cited Moron and Gouirand 2003, among others). **Tag: R1.1, R2.2**

**3.** The authors analyze the effect of separating the ENSO events into CP and EP events. I think, given the nature of the paper, it would be important to analyze what is the effect of the ENSO magnitude, i.e. consider strong (Nino3.4>1.5/2SD) and moderate (1SD>Nino3.4>1.5/2SD) ENSO events separately. I think the current length of the paper would allow for the addition of this analysis.

We have now calculated these additional composites based on event strengths as suggested. Additionally, we added an 'extreme' category (see Fig. R1 and caption below). Very roughly, the stronger ENSO events appear to be associated with stronger anomalies over the North Atlantic. However, it is difficult to be certain about this for a number of reasons:

**a.** Sample sizes are smaller after these separations and so statistical significance is severely affected in many cases. (please see also somewhat related point R1.2 above)

**b.** There is only one extreme event for extreme ND or JF La Nina.

**c.** The composite for extreme JF El Nino (Fig. R1l, 4 events) appears to be fundamentally different. This is a well-known result shown in Toniazzo and Scaife (2006, https://agupubs.onlinelibrary.wiley.com/doi/10.1029/2006GL027881)

[Figure]

*Fig. R1 SLP composites for El Niño and La Niña events during 1870-2014 in ND (first and second columns) and JF (third and fourth columns) means. Each row of these panels corresponds to the first row of Fig. 5 except here there is further separation according to the ENSO amplitude in terms of the standard deviation of the Niño2.4 index: moderate (1 SD ≤ |Niño3.4| < 1.5 SD), strong (| Niño3.4| ≥ 1.5 SD), and extreme (|Niño3.4| ≥ 2.0 SD) (Note that the 'strong' category also includes the 'extreme' events). There is only one extreme La Niña event each for ND and JF. Red (blue) contours indicate positive (negative) anomalies, with gray shading indicating the 5% significance level for a two-tailed t test. In first and second rows: contour interval = 1hPa, in third row: contour interval = 2hPa.*

We decided not to include this result in the manuscript because we believe these additional calculations do not do justice to the nonlinearity topic nor add to previous findings (such as those cited in R2.1 above, especially Trascasa-Castro et al. 2019). As these authors and you have indicated, which we agree, the results concerning nonlinearity in ENSO teleconnections based on only the reanalysis data are very uncertain (perhaps this shouldn't be a surprise as even the certainty in the linear composites from reanalysis is being questioned).

On the other hand, in addressing your points R2.25 and R2.26 below about the asymmetrical parts in terms of the El Nino+La Nina composites, we decided that the result is a warranted addition to the main text and figure. We recognise that there are a number of other factors that may affect nonlinearity which we cannot deal with in the manuscript (amplitudes of ENSO being one of them) and these are mentioned as a caveat in Sect 3.2. Please also see R2.25 and R2.26 below.

**R2.3**

**4.** Section 3.3 discusses different methods to compute confidence intervals. However, the conclusion of this section is rather "boring". Figures 7 and 8 look almost the same for all the different methods employed. I understand that the reason why the authors have decided to put these figures in the main text is to show that they are actually very similar. Nevertheless, I would recommend putting these two figures as an appendix as they do not provide a lot of quantitative information, but they are a justification for the methods employed in the first part of the study.

The original complete set of Figs. 7, 8 are now moved to the appendix figures. Now, only the confidence intervals for the JF El Nino composite are shown in the main figure. **Tag: R2.4**

**Specific comments (line by line):**

**Line 10:** What does "some confidence in the signs" means here? Is there a more objective way to state this?

The description is modified to "... we can have confidence (at the 5% significance level)...".

This statement is based on the results in Sect. 3.3 as well as applying the standard t-test for the composites in Sect. 3.1.

Please see Abstract. **Tag: R2.5**

**Lines 11-12:** Is the lower confidence in the amplitude due to internal North Atlantic variability or due to ENSO diversity (asymmetry and nonlinearity)?

It should be due to the internal North Atlantic SLP variability. Although the ENSO events exhibit variability/diversity, the average of the selected ENSO events (i.e., the composite) from the reanalysis appears to be very stable (do not vary among the bootstrap composites). This is the main thesis of Deser et al. (2017) and our study. Please also refer to the Abstract, Fig. 5 and related text in the cited Deser et al. (2017).

Added explanation in the second paragraph of Introduction. **Tag: R2.6**

**Line 20:** All teleconnections are uncertain in amplitude due to internal variability no? Maybe one should say if this amplitude uncertainty is similar or smaller than in the North Atlantic?

The description is modified according to your suggestion. Please see paragraph 2 in Introduction. **Tag: R2.7**

**Line 22-23:** What do you mean by "detectable response"? Do you mean that the sign of the teleconnection is uncertain?

Sorry for the confusion. We didn't have a very precise meaning for "detectable" other than to mean the composites (both in sign and amplitude) calculated from the reanalysis data. We changed "detectable" to "observed". Second paragraph in Introduction. **Tag: R2.8**

**Line 25:** In my opinion, the ND pattern is not similar to the NAO pattern, as one center is much stronger than the other, so it is not just a shift, but more similar to a monopole that resembles the EA pattern (e.g. Wollings et al., 2010)

Woollings, T., Hannachi, A., & Hoskins, B. (2010). Variability of the North Atlantic eddy-driven jet stream. *Quarterly Journal of the Royal Meteorological Society*, *136*(649), 856–868. https://doi.org/10.1002/qj.625

That part of the sentence is modified to "The Nov-Dec teleconnection resembles the East Atlantic pattern;...". Please see the third paragraph in Introduction. **Tag: R2.9**

**Line 26:** I think we cannot say that the teleconnection is "the NAO pattern" or NAO-like pattern, but that it projects onto the NAO pattern. Related to this point, the authors might refer to the paper by Mezzina et al. (2020a,b).

Mezzina, B., García-Serrano, J., Bladé, I., & Kucharski, F. (2020). Dynamics of the ENSO Teleconnection and NAO Variability in the North Atlantic–European Late Winter. *Journal of Climate*, *33*(3), 907–923. https://doi.org/10.1175/JCLI-D-19-0192.1

Mezzina, B., García-Serrano, J., Bladé, I., Palmeiro, F. M., Batté, L., Ardilouze, C., … Gualdi, S. (2020). Multi-model assessment of the late-winter extra-tropical response to El Niño and La Niña. Climate Dynamics, 1, 3. https://doi.org/10.1007/s00382-020-05415-y

This part of the sentence is modified to "... while the Jan-Feb teleconnection projects onto the NAO pattern (Mezzina et al. 2020)." Please see the third paragraph in Introduction. **Tag: R2.10**

**Line 30:** Maybe you could add the paper by Jimenez-Esteve and Domeisen (2018) where the tropospheric mechanisms in late winter are analyzed together under the influence of the stratospheric pathway.

Jiménez-Esteve, B., & Domeisen, D. I. V. (2018). The tropospheric pathway of the ENSO-North Atlantic teleconnection. *Journal of Climate*, *31*(11), 4563–4584. https://doi.org/10.1175/JCLI-D-17-0716.1

Added. See third paragraph in Introduction. **Tag: R2.11**

**Line 34:** Actually, Hardimann et al. (2019) also find that this Rossby wave train is important throughout the season, mainly for strong EN events, but it is more muted for moderate EN or LN events when the stratospheric pathway is active in late winter.

Hardiman, S. C., Dunstone, N. J., Scaife, A. A., Smith, D. M., Ineson, S., Lim, J., & Fereday, D. (2019). The Impact of Strong El Niño and La Niña events on the North Atlantic. *Geophysical Research Letters*, *46*(5), 2874–2883. https://doi.org/10.1029/2018GL081776

This point is recognised in the previous paragraph (please see the penultimate sentence in the third paragraph of Introduction). We now added reference to Hardiman et al., 2019. **Tag: R2.12**

Paragraph 4 in the Introduction is describing the teleconnection in Nov-Dec specifically, with a number of recent studies reporting interbasin effects from western tropical Atlantic and the Indian Ocean, etc.

**Line 56:** "larger" than what?

We have changed the description. Our domain includes the midlatitudes and encloses both centres of anomalies in the North Atlantic, in contrast to Deser et al. (2017) who used a polar area. See last paragraph in Introduction. **Tag: R2.13**

**Line 58:** A "more optimistic" perspective compared to? DESER17? I am not sure if "optimistic" is the right term to use here as optimism is subjective and not objective science.

The description/wording is modified. Please see the last sentence in Introduction and the second paragraph in Concluding remarks. Please also see our reply to your related comment below in R2.28. **Tag: R2.14**

**Line 66:** Why are the two citations after a full stop?

The typo is corrected. **Tag: R2.15**

**Line 68-69:** For what months or running mean is the Nino3.4 averaged?

This information is described in the first paragraph of Sect 3.1. Since our analyses involve DJF, ND and JF, (also for periods 1920-2014 or 1870-2014) we feel that it is more convenient to describe the selection of the events closer to where the results are reported. We added a sentence in Sect. 2.2 now ("Table 1 lists the selected ENSO events for calculating the DJF, Nov-Dec and Jan-Feb SLP composites, further explanation is given in the beginning of Sect. 3.1"). **Tag: R2.16**

**Line 72:** How many years do you draw every time from the $E\_0$ and $L\_0$ sets every time (2000 times)? In your expression of $C^*$, $E^*$ and $L^*$ are averaged before computing the difference, is this correct? Please clarify.

The bootstrap sample size is the same as the original/observed sample size (same approach in Deser et al. 2017). This is an appropriate choice for investigating the uncertainty of the observed samples with their specific finite sizes. In an added part in Sect 3.1 we also investigate the effect of sample size (from R1.2). A sentence clarifying this point is added in Sect 2.2. **Tag: R2.17**

Yes, $C^*$ is calculated as the average of $E^*$ minus the average of $L^*$ (not different from the common way a composite is calculated). This is already given in the equations with overbar terms, with the meaning of the overbar described. Note that normally we would probably not need to use the notations/equations for the explanation in Sect. 2.2. But in this study we need the notations for explaining the calculations of the different types of confidence intervals in Sect. 3.3.

**Lines 89-98:** After reading this subsection, I understand the meaning of what is represented in the Taylor diagram, i.e the amplitude and spatial correlations, but I struggle to understand what exactly each point in the Taylor diagram represents. For example, what do the $\| \ \|_2$ operators symbolize? A better explanation of how the spatial information is averaged in this diagram would be desired.

The symbol means Euclidean norm or L2-norm (ie. the square root of sum of squares). We write out the equations in expanded form now, so the compact form is not needed anymore. The meaning of the polar coordinates $(r, \alpha)$ of each dot is described in words and equations. Please see Sect. 2.4. **Tag: R2.18**

**Line 103:** Why using different definitions of ENSO depending on the period analyzed? I think the common approach is defining ENSO events based on the NDJ or ONDJF Nino3.4 mean and then use the same years for the 2-months composites (ND or JF). I am aware this will not lead to different results, but I think it complicates the methodology more than necessary.

We added an explanation in the beginning of Sect. 3.1. **Tag: R2.19**

One reason we would use NDJ to define Nino3.4 for selecting DJF extratratropics anomalies is to account for a delay (through both the tropospheric and stratospheric pathways) to the extratropical surface anomalies (Deser et al. 2017 use same approach). Following this reasoning for the one-month lag, we use ON Nino3.4 for selecting ND SLP anomalies, and DJ Nino3.4 for selecting JF anomalies. This may not be a widely-used definition of ENSO events per se (as compared to ONDJF, for example). However, as you mentioned this does not affect the results, mainly because ENSO related anomalies often persist from about Oct to Feb. Looking at the years selected in Table 1, we see that in fact NDJ and DJ events are identical. And the ON events are mostly the same ones as well. Only one or two years when the Nino3.4 values are close to the threshold then there are differences.

**Lines 116-117:** It is inaccurate to say that the meridional gradient is opposite for ND than JF, and just say that the centers are shifted, while as you mention in the intro, this pattern, although maybe projecting weakly on the positive NAO, is more closely related to the EA pattern, which usually leads to a strengthening of the North Atlantic jet stream (Woollings et al., 2010).

We deleted "... shows a meridional gradient in the North Atlantic opposite of that for JF, and the centers located south of those for JF." And now simply describe it as resembling the East Atlantic pattern, consistent with Introduction. **Tag: R2.20**

**Figure 3:** Maybe for consistency with Figure 2 you could also show the 5[th] and 95[th] percentile maps when using only the Icelandic low box.

These maps are now added to Fig. 3, and the description at the end of the 4th paragraph in Sect. 3.1 is modified to include the new information. **Tag: R2.21**

**Lines 143-147:** Including a larger area does not decrease the uncertainty in the polar regions (>60degN), but because the uncertainty is much lower over the Azores anticyclone, this compensates the uncertainty of the Icelandic low center when both are averaged in your spatial correlation number.

Yes, you are right. The metric is based on the chosen domain, i.e. it includes the more southern area which has better statistical significance. We deleted this part "... as well as including the mid-latitude North Atlantic in the analysis, ..." **Tag: R2.22**

**Lines 161-170:** I am not very convinced why the authors need to prove that EN is different than LN? Isn't this obvious? What one could test here is if the teleconnection pattern is different (asymmetry). This could be done by multiplying the SLP pattern for LN years by -1, and then perform the analysis that the authors propose here.

We agree that the permutation tests are not strictly necessary after the tests in Figs. 2, A2. However, we think that the test and result are interesting, and would like to use the opportunity to show another method for resampling. We have moved this part to Appendix A and refer to it near the end of Sect. 3.1. **Tag: R2.23**

(Please see reply to R2.25 and R2.26 below regarding the test for asymmetry).

**Lines 171-173:** I do not understand where the 30/2000, 16/2000 and 11/2000 values come from? I also assume in Figure 5 you are using the 1870-2014 period as you refer to figure A1, but this is not mentioned in the text. Could you please clarify?

The numerators of these fractions are the number of red dots in the respective panels (i.e., the red dots are those bootstrap composites having amplitudes at least as large as the amplitudes of the observed composites). And 2000 is just the total number of bootstrap samples. Additional explanation is given now. Please see paragraph 2 in Appendix A. The period "1870-2014" is added now to the figure caption. **Tag: R2.24**

Note that this part is moved to Appendix A per reply R2.23 above.

**Line 187:** I do not agree with the statement that there is no sign of asymmetry between EN and LN response. I can see a more zonally extended NAO-like dipole response for JF LN or a much stronger deepening of the Aleutian low in the North Pacific for EN. The asymmetry seems to be more evident in late winter. This is asymmetry from my understanding. Alternatively, one could do bootstrapping to analyze the asymmetry similar as in Figures 1 and 2 but using the EN+LN composite instead.

We wrote "...do not reveal any sign asymmetry" - meaning there is no asymmetry in signs (not considering asymmetry/nonlinearity in amplitudes) of the SLP anomalies. Also, the analyses using Taylor diagrams are based on the domain as a whole, not specific parts of the domain. This sentence has been edited for clarity. Please see the second paragraph in Sect. 3.2. Additionally, we previously mentioned in the same paragraph the 'apparent' asymmetry in some locations. **Tag: R2.25**.

Regarding testing the EN+LN composites... We had previously performed this analysis but were reluctant to include the result partly because nonlinearity is not our main focus and partly because we think this is a challenging problem that we cannot address comprehensively here. However, we are now persuaded to include this result. It is described in a new paragraph in Sect. 3.2 together with a new main figure. Some caveats on the limitations of the result are included. **Tag: R2.25**

Similar to the papers you also referred to below in R2.26, we could not show statistically significant nonlinearity using the reanalysis data. We do not claim that there is no nonlinearity per se. Only that our specific analyses (domains, variables, ENSO definition, limited reanalysis events) cannot show it. These factors are mentioned as the caveats to the added result. **Tag: R2.25**

**Line 209:** If the purpose of this paper is uncertainty, I think the authors should try to see if the uncertainties in the nonlinearity/asymmetry are significant or not (see my previous comment).

Previous studies identifying nonlinearities (e.g., Trascasa-Castro 2019; and Jimenez-Esteve and Domeisen, 2020) have shown that those are only statistically significant in model simulations where a large sample of strong ENSO events can be simulated.

Please see our reply to R2.25 above. What we have found is consistent with your summary here. **Tag: R2.26**

**Lines 259-260:** As I mentioned before I wouldn't call the early-winter response a dipole response.

The word "dipole" is deleted. Please see second paragraph in Sect. 4. **Tag: R2.27**

**Lines 264-265:** What is the thing that makes your view more optimistic than previous studies? This optimistic view that you claim needs a more clear justification. Most previous studies acknowledge that internal variability in the North Atlantic is large and those signals are nor always are robust as one would like, but that does not mean that other papers are less optimistic. Nevertheless, optimism is a subjective vision and I think science should try to be objective

We have modified our wording and description, please see also reply to R2.14 above. **Tag: R2.28**

By "more optimistic" we had meant to argue that we can have confidence in the patterns of the teleconnection in the North Atlantic, which is not the message one might get from Deser et al. (2017). See in particular their Figs. 5b, 6, and conclusions:

"Even with nearly 100 years of data comprising 18 El Niño and 14 La Niña events, we have shown that the observed extratropical NH SLP response to ENSO in boreal winter (DJF) is subject to considerable uncertainty in pattern and amplitude...",

"The observed SLP composite shows a robust ENSO response over the North Pacific and North America...",

"Other regions, such as the Arctic, North Atlantic, and Europe, show a larger range of patterns, amplitudes, and statistical significance across the bootstrapped samples."

[revised manuscript text omitted]